



# Inverse modeling of SO₂ and NOₓ emissions over China using multi-sensor satellite data: 2. Downscaling techniques for air quality analysis and forecasts

Yi Wang[1], Jun Wang[1,2], Meng Zhou[1], Daven K. Henze[3], Cui Ge[2,4], Wei Wang[5]

[1]Interdisciplinary Graduate Program in Informatics, The University of Iowa, Iowa City, IA 52242, USA
[2]Department of Chemical and Biochemical Engineering, and Center for Global & Regional Environmental Research, The University of Iowa, Iowa City, IA 52242, USA
[3]Department of Mechanical Engineering, University of Colorado, Boulder, CO 80309, USA
[4]South Coast Air Quality Management District, Diamond Bar, CA 91765, USA
[5]China National Environmental Monitoring Center, Beijing 100012, China

*Correspondence to*: Jun Wang (jun-wang-1@uiowa.edu) and Yi Wang (yi-wang-4@uiowa.edu)

**Abstract.** Top-down emissions estimates provide valuable up-to-date information on pollution sources; however,
the computational effort involved with developing these emissions often requires them to be estimated at resolutions that are much coarser than is necessary for regional air-quality forecasting. This work thus introduces several approaches to downscaling coarse-resolution (2°x2.5°) posterior SO₂ and NOₓ emissions (derived through inverse modeling in Part I of this study) for improving air quality assessment and forecasts over China in October 2013. The SO₂ and NOₓ emission inverse modeling was conducted at the 2°x2.5° resolution in Part I to save
computational time. The prior emission inventory (MIX) as well as the posterior GEOS-Chem simulations of surface SO₂ and NO₂ concentrations at this resolution underestimate observed hot spots, which is called the Coarse-Grid Smearing (CGS) effect. To mitigate the CGS effect, four methods are developed: (a) downscale 2°x2.5° GEOS-Chem surface SO₂ and NO₂ concentrations to the resolution of 0.25°x0.3125° through a Dynamic Downscaling Concentration (MIX-DDC) approach, which assumes that the 0.25°x0.3125° simulation using the
prior MIX emissions has the correct spatial distribution of SO₂ and NO₂ concentrations but a systematic bias; (b) downscale surface NO₂ simulations at 2°x2.5° to 0.05°x0.05° according to the spatial distribution of Visible Infrared Imaging Radiometer Suite (VIIRS) Nighttime Light observations (e.g., NL-DC approach) based on correlation between VIIRS NL intensity with TROPOMI NO₂ observations; (c) Downscale posterior Emissions (DE) of SO₂ and NOₓ to 0.25°x0.3125° with the assumption that the prior fine-resolution MIX inventory has the
correct spatial distribution (e.g., MIX-DE approach); and (d) downscale posterior NOₓ emissions using VIIRS NL observations (e.g., NL-DE approach). Numerical experiments reveal that: (a) using the MIX-DDC approach,





posterior $SO_2$ and $NO_2$ simulations improve compared to the corresponding MIX prior simulations with normalized centered root mean square error (NCRMSE) decreases of 63.7% and 30.2%, respectively; (b) the $NO_2$ simulation has an NCRMSE that is 17.9% smaller than the prior $NO_2$ simulation when they are both downscaled through NL_DC, and NL_DC is able to better mitigate the CGS effect than MIX-DDC; (c) the simulation at 0.25°x0.3125° using the MIX-DE approach has NCRMSEs that are 58.8% and 14.7% smaller than the prior 0.25°x0.3125° MIX simulation for surface $SO_2$ and $NO_2$ concentrations, respectively, but the RMSE from the MIX-DE posterior simulation is slightly larger than that from the MIX-DDC posterior simulation for both $SO_2$ and $NO_2$; (d) the NL-DE posterior $NO_2$ simulation also improves on the prior MIX simulation at 0.25°x0.3125°, but it is worse than the MIX-DE posterior simulation; (e) in terms of evaluating the downscaled $SO_2$ and $NO_2$ simulations simultaneously, using the posterior $SO_2$ and $NO_x$ emissions from joint inverse modeling of both species is better than only using one ($SO_2$ or $NO_x$) emissions from corresponding single-species inverse modeling and is similar to using the posterior emissions for both $SO_2$ and inventories from single-species inverse modeling. Forecasts of surface concentrations for November 2013 using the posterior emissions obtained by applying the posterior MIX-DE emissions for October 2013 with the monthly variation information derived from the prior MIX emission inventory show (a) the improvements of forecasting surface $SO_2$ concentrations through MIX-DE and MIX-DDC are comparable; (b) for $NO_2$ forecast, MIX-DE show larger improvement than NL-DE and MIX-DDC; (c) NL-DC is able to better decrease the CGS effect than MIX-DE, but shows larger NCRMSE. Overall, for practical forecasting of air quality, it is recommended to use satellite-based observation already available from the last month to jointly constrain $SO_2$ and $NO_2$ emissions at coarser resolution and then downscale these posterior emissions at finer spatial resolution suitable for regional air quality model for the present month.

## 1. Introduction

Simulations and forecasts of surface $SO_2$ and $NO_2$ concentrations are important for studying their impacts on air quality and public health (Ghozikali et al., 2016). Their accuracy depends not only on precise meteorological fields and chemically correct air pollution simulations (Gao et al., 2016;Ge et al., 2017) but also on the fidelity of the emissions used in the latter. For the same region and time, different emission inventories can lead to differences of up to 100% for surface $SO_2$ and $NO_2$ simulations (Wang et al., 2016b). Additionally, model resolution also plays an important role (Kharol et al., 2017), as the simulated concentration of these short-lived species only represents the average of a grid cell in which the high concentrations of $SO_2$ and $NO_2$ from source regions and (or) strong spatial variation of these species are smeared out. This is called the Coarse-Grid Smearing (CGS)



effect, and it depends on the species lifetimes, the spatial distribution of emissions, and the model (and inventory) resolution. The lifetime for $SO_2$ in the troposphere is less than one day in the summer and one or two days in winter (Lee et al., 2011) and it is several hours for $NO_2$ (Lin et al., 2010); their smearing length scales (Palmer et al., 2003) are of order of 100 km (Lee et al., 2011;Martin et al., 2003). Xing et al. (2015) showed that surface

$SO_2$ and $NO_2$ concentrations from the Weather Research and Forecasting (WRF)– Community Multi-scale Air Quality (CMAQ) simulations at 108 km x108 km resolution were underestimated when validated against urban network observations and overestimated relative to rural networks.

Obtaining accurate and timely emission estimates can be challenging. The bottom-up approach, which integrates

activity data and emission factors, is widely used to generate inventories (Li et al., 2017b;Janssens-Maenhout et al., 2015;Kurokawa et al., 2013). These bottom-up emissions have uncertainties larger than 30% at the regional scale for both $SO_2$ and $NO_x$ over China (Kurokawa et al., 2013;Li et al., 2017b). When used to simulate air quality with a Chemical Transport Model (CTM), these emission estimates are gridded to regular grid cells (of ~1° or finer) through locations of major manufacturing facilities and power plants and proxy data such as population

distributions and road networks (Zheng et al., 2017;Streets et al., 2003). Consequently, uncertainties of emissions estimates at the grid-cell scale are larger than country scale. Moreover, bottom-up inventories usually have a time lag of at least one year, as it takes time to collect all the data required to generate them (Liu et al., 2018). Outdated emission inventories increase the uncertainty of simulations and forecasts, especially for China where emissions change quickly due to rapid economic development and implementation of emission control policies (Zheng et

al., 2018;Wang et al., 2016b).

Over the past two decades, many satellites have provided Vertical Column Density (VCD) data of $SO_2$ and $NO_2$ and Aerosol Optical depth (AOD) retrievals globally; these data have been used to constrain emissions estimates with the following approaches at various spatial resolutions. The mass balance approach (Lee et al., 2011;Martin

et al., 2003;Koukouli et al., 2018) and the finite difference mass balance method (Lamsal et al., 2011) were developed to use VCD retrievals of $SO_2$ and $NO_2$ from Global Ozone Monitoring Experiment (GOME), GOME-2, SCanning Imaging Absorption SpectroMeter for Atmospheric CHartographY (SCIAMCHY), Ozone Monitoring Instrument (OMI), and Ozone Mapper and Profiler Suite (OMPS) to constrain $SO_2$ and $NO_x$ emissions at spatial resolutions in the range of 25 km to 250 km. The accuracy, however, decreases as spatial resolution

becomes finer (Turner et al., 2012), as transport is not explicitly accounted for in these approaches. The emission strength of $SO_2$ point sources that are larger than 30 kt per year can be estimated through a linear regression



between OMI VCDs and emission strength (Fioletov et al., 2016), and the approach was used to build a global SO₂ emission inventory at 0.1°x0.1° (Liu et al., 2018). Advanced data assimilation approaches including the four-dimension variational data assimilation (4D-Var) (Qu et al., 2019a;Qu et al., 2019b;Qu et al., 2017;Wang et al.,

2016b;Wang et al., 2019;Wang et al., 2012;Xu et al., 2013;Kurokawa et al., 2009) and the Ensemble Kalman Filter (EnKF) approach (Miyazaki et al., 2012;Miyazaki et al., 2017) were developed to use satellite SO₂ and NO₂ columns densities and AOD retrievals to constrain emissions at low spatial resolutions (>50 km) as these approaches are computation-intensive. Some variations of the 4D-Var and Kalman filter approaches were developed to save computational time at the expense of accuracy or temporal resolution (Qu et al., 2017;Kong et

al., 2019;Mijling and van der A, 2012;Ding et al., 2015).

The mismatch among the resolutions of emission inventories, CTMs, and satellite observations has prompted previous development of downscaling methods. For example, the popular OMI has a footprint size of 13 km x 24 km at nadir and 26 km x 128 km at the swath edge that is too coarse to capture urban NO₂ plume without

oversampling. Consequently, a spatial weighting kernel derived from the CMAQ simulation at finer spatial resolution was developed to downscale OMI NO₂ retrievals to 1.33 km x 1.33 km (Kim et al., 2018;Kim et al., 2016;Goldberg et al., 2017). The resulting high-spatial-resolution OMI NO₂ data was further applied to constrain emissions, which showed an underestimate in the bottom-up NOₓ inventories in Seoul, South Korea during the Korea-United States Air Quality Study (KORUS-AQ) (Goldberg et al., 2019). In cases that model grid cells are

larger than satellite footprints, Lamsal et al. (2008) applied the ratio between local OMI NO₂ column to mean OMI field over a 2°x2.5° GEOS-Chem grid cell to derive local surface-VCD scaling factors, which were used to infer improved surface NO₂ concentrations. An inverse distance weighting technique was applied to interpolate emissions and initial and boundary species conditions from coarse resolution to fine resolution for nested CTM simulations (Yahya et al., 2017;Yahya et al., 2016;Hong et al., 2017), but it was not able to capture hot spots in

the downscaled fields.

The CGS effect, combined with the sharp spatial variations of surface SO₂ and NO₂ concentrations, introduces challenges when comparing model simulations with in situ observations.  Wang et al. (2016b) showed the improvement of using posterior SO₂ emissions constrained by OMI SO₂ to simulate surface SO₂ concentrations

at a resolution of 2°x2.5°.  However, this was illustrated for a rural site that is ~100 km away from Beijing's urban center, and there are no strong SO₂ sources around it, which means the CGS effect is minimal at this site. Kharol et al. (2015) and Kharol et al. (2017) found that surface SO₂ and NO₂ concentrations derived through scaling OMI




SO₂ and NO₂ VCDs with vertical profiles from a CTM at a resolution of 0.1°x0.1° are a factor two smaller than US EPA in situ observations. These underestimations are partly ascribed to the CGS effect, although uncertainty in vertical profiles also plays a role (Kharol et al., 2015;Kharol et al., 2017;Bechle et al., 2013). They further showed that the underestimation decreases significantly when in situ observations are converted to represent the averages of larger areas through a linear regression function which is built by comparing simulations of SO₂ between two spatial resolutions of 2.5x2.5 km² and 30x30 km².

This paper, as the second of a two-part study, aims at using SO₂ and NOₓ emissions constrained by OMPS SO₂ and NO₂ retrievals through 4D-Var (which is presented in part I, i.e. Wang et al. (2019)) to improve air quality forecasts. Since the emission inventories in Part I are derived at the 2°x2.5° resolution to save computational resources, the focus here is to develop novel methods to downscale coarse-resolution emission inventories or simulation results to generate fine-resolution surface SO₂ and NO₂ concentrations and evaluate them from an air quality forecasting point of view. High-resolution bottom-up emission inventories and Visible Infrared Imaging Radiometer Suite (VIIRS) nighttime lights contain geospatial information (such as roads, location of power plants, and residential areas) in fine spatial resolution for downscaling coarser-resolution anthropogenic emissions. Indeed, VIIRS nighttime light observations are shown to be good indicators of socioeconomic parameters including urbanization, economic activity, population (Bennett and Smith, 2017), road density (Levin and Zhang, 2017), and have been used to map CO₂ emissions (Ou et al., 2015) and derive surface PM₂.₅ concentrations at nighttime (Wang et al., 2016a). Thus, it should also be promising to build relationships between VIIRS nighttime lights and both NO₂ in the atmosphere and NOₓ emissions, which will be assessed here for its application in downscaling surface NO₂ concentrations and NOₓ emissions.

We introduce data in Sect. 2. Section 3 presents the models for simulations and forecasts of surface SO₂ and NO₂, and the downscaling approaches. The improvements in the simulations and forecasts through various downscaling methods are provided in Sect. 4. Discussions of implications of the results and conclusions are followed in Sect. 5.

## 2. Data

### 2.1 In situ surface SO₂ and NO₂



We use in situ surface $SO_2$ and $NO_2$ measurements from the China National Environmental Monitoring Center for model evaluation. $SO_2$ and $NO_2$ are measured by various commercial instruments using the ultraviolet fluorescence method and the chemiluminescence method, respectively (Zhang and Cao, 2015). In the chemiluminescence method $NO_2$ observations are obtained by measuring NO from decomposed $NO_2$. This can

result in a positive bias because $NO_z$ (all compounds that are products of the atmospheric oxidation of $NO_x$) will be also reduced to NO. Steinbacher et al. (2007) showed that the ratio of $NO_2$ to $NO_z$ ($r_{NO2}$) depends on the distance that $NO_2$ plumes transport from the source. In other words, the longer the distance, the more the potential for oxidation of $NO_2$, hence the smaller $r_{NO2}$; only 43% - 76% and 70% - 83% of real $NO_2$ contribute to the measured value $(NO_2)_m$ for rural and urban sites, respectively (Steinbacher et al., 2007). For this study, as

observational sites are in cities, a maximum value of 0.83 is used to convert $(NO_2)_m$ measurements to the $NO_2$ concentrations, which is subsequently used for evaluating the model results. Additionally, we also test values for $r_{NO2}$ in the range of 0.7 to 1.0.

## 2.2 VIIRS data for artificial light

The VIIRS on board National Polar-orbiting Partnership (Suomi-NPP) satellite was launched on 28 October 2011,

and its Day/Night Band (DNB) provides observations of nighttime lights with a spatial resolution of 750 m (Miller et al., 2013). Here, we use the VIIRS nighttime lights product that has excluded background noise, solar and lunar contamination and has screened out the data degraded by cloud cover and features unrelated to electric lighting (Elvidge et al., 2017). This product is regridded to 0.05°x0.05° for October 2013 and to 0.05°x0.05° and 0.25°x0.25° for April 2018.

## 170  2.3 TROPOMI NO$_2$ tropospheric VCD

The TROPOspheric Monitoring Instrument (TROPOMI) on board Sentinel-5 Precursor was launched on 13 October 2017, with a nadir footprint of 7x3.5 km², which is finer than that of all its predecessors. The TROPOMI $NO_2$ tropospheric VCDs from Royal Netherlands Meteorological Institute (KNMI) were retrieved using a Differential Optical Absorption Spectroscopy (DOAS) algorithm and validated with Pandora $NO_2$ retrievals

(Griffin et al., 2019). We grid the product to the 0.05°x0.05° resolution for April 2018 to investigate the relationship between VIIRS nighttime lights and $NO_2$ tropospheric VCDs.

## 2.4 MIX emission inventory





MIX (Li et al., 2017a) is a mosaic of Asian anthropogenic monthly emissions developed for the years 2008 and 2010 to support the Model Inter-Comparison Study for Asia and the Task Force on Hemispheric Transport of Air
Pollution. $SO_2$, $NO_x$, and $NH_3$ emissions in MIX come from the Regional Emission inventory in ASia version 2.1 (REAS2.1) (Kurokawa et al., 2013), replaced by the MEIC $SO_2$ and $NO_x$ and the PKU $NH_3$ (Huang et al., 2012) for mainland China, the ANL (Lu et al., 2011;Lu and Streets, 2012) $SO_2$ and $NO_x$ of some source sectors for India, and the CAPSS $SO_2$ and $NO_x$ for the Republic of Korea (Li et al., 2017b). In spite of variations among spatial resolutions of these emission inventories, they are regridded to 0.25°x0.25° to form the MIX emissions inventory
(Li et al., 2017a). In our study, not only is MIX used in the posterior simulations and forecasts, but it also provides information for downscaling the posterior emission inventories from Part I (as in Wang et al. (2019)).

## 3. Methods

### 3.1 GEOS-Chem and configuration

The CTM used for the simulations and forecasts of surface $SO_2$ and $NO_2$ concentrations is GEOS-Chem version
12.0.0 (GCv12.0.0), which is driven by GEOS-FP meteorological fields from GMAO. Horizontal resolutions are set as 2°x2.5°, the same one of posterior emissions from Part I (as in Wang et al. (2019)), and 0.25°x0.3125°, which is the finest resolution available for this version of GEOS-Chem, to investigate the impacts of downscaling on simulations and forecasts. There are 47 vertical layers, the lowest one of which represents surface concentrations validated against in situ observations. We use the MIX 2010 emissions for October 2013 prior
simulations as well as November 2013 prior forecasts. Posterior $SO_2$ and $NO_x$ emissions for October 2013 from Part I, i.e. Wang et al. (2019), are used for October 2013 simulations and November 2013 forecasts at 2°x2.5° resolution, but need be downscaled for 0.25°x0.3125° simulations, as described in Sect. 3.3.

It is worth noting that the GEOS-Chem adjoint model (Henze et al., 2007) used in Part I, i.e. Wang et al. (2019),
is v35m, which is developed based on GEOS-Chem version 8.2.1, updated through version 9. Here we use GCv12.0.0 rather than GC adjoint v35m to investigate if the model-dependent posterior emission inventory can be applied to other models to improve simulations and forecasts. With the same MIX emissions used, GCv12.0.0 surface $SO_2$ and $NO_2$ concentrations are in general larger than that from v35m, with differences of up to 15 µg m$^{-3}$ for $SO_2$ and 10 µg m$^{-3}$ for $NO_2$ (Fig. 1). The difference is due to differences in chemical mechanism and boundary
layer parameterization schemes between the two models. Therefore, by using two different versions of GC, we





can study the degree to which the posterior emissions derived from one model (in this case global, with coarser resolution) can be applied for another (here a regional model with finer resolution).

## 3.2 Downscaling GEOS-Chem surface concentrations

GEOS-Chem surface $SO_2$ and $NO_2$ concentrations at a resolution of 2°x2.5° are not expected to be able to capture
hot spots due to the CGS effect, and thus we aim to downscale them to finer resolutions. The prior emissions are MIX for October and November 2010. The posterior emissions are from separate inverse emission estimates in Part I (e.g., E-$SO_2$ and E-$NO_2$ experiments as described in Wang et al. (2019)), unless it is specifically stated. The downscaling methods here should be distinguished from interpolation approaches to simply increasing spatial resolutions.

### 3.2.1 Downscaling concentrations with MIX simulations

With the assumption that surface concentrations of GEOS-Chem simulations using outdated emissions have correct spatial distributions at fine scales but systemic bias at coarse scales , we use 0.25°x0.3125° prior surface concentration patterns to downscale both prior and posterior 2°x2.5° simulations of surface species concentrations as shown in Eq. (1).

$$C_{f,i}^{MIX-DC} = C_c \times \frac{C_{f,i}^{pri}}{1/64 \times \sum_{i=1}^{64} C_{f,i}^{pri}} \quad (1)$$

A coarse 2°x2.5° grid cell consists of 64 (8x8) fine 0.25°x0.3125° grid cells, and $C_{f,i}^{pri}$ represents the MIX prior simulation of surface concentrations from the i$^{th}$ 0.25°x0.3125° grid cell within a 2°x2.5° grid cell. Thus, $1/64 \times \sum_{i=1}^{64} C_{f,i}^{pri}$ is the mean 0.25°x0.3125° simulation in a 2°x2.5° grid cell, and $\frac{C_{f,i}^{pri}}{1/64 \times \sum_{i=1}^{64} C_{f,i}^{pri}}$ is the fine-to-coarse ratio, which multiples the 2°x2.5° surface concentration, $C_c$, to obtain the downscaled result $C_{f,i}^{MIX-DC}$. This
approach is titled Dynamic Downscaling Concentration with MIX simulation (MIX-DDC). Here, dynamic downscaling means the application of fine-scale model concentrations to downscale coarse resolution concentrations.

### 3.2.2 Downscaling concentrations with nighttime lights

The VIIRS nighttime lights product at a resolution of 0.05°x0.05° is used to downscale GEOS-Chem simulations
of surface $NO_2$ due to its high spatial resolution and strong correlation with population distribution (Bennett and Smith, 2017) as well as $NO_2$ VCDs. Figures 2a-b show the spatial distributions of VIIRS nighttime lights and



TROPOMI $NO_2$ tropospheric VCDs over China, and it is not surprising that both nighttime lights and $NO_2$ hot spots are mainly over metropolises. Figure 2c shows strong linear correlation between the logarithm of VIIRS nighttime lights and TROPOMI $NO_2$ tropospheric VCDs at a resolution of 0.05°x0.05°, and this relationship is used to downscale as shown in Eq. (2) and (3).

$$W_i = \ln(V_i) - \ln(0.1) \quad (2)$$

$$C_{f,i}^{NL-DC} = C_c \times \frac{W_i}{\overline{W}} \quad (3)$$

$V_i$ represents the $i^{th}$ VIIRS 0.05°x0.05° nighttime light in a 2°x2.5° grid cell and all nighttime lights less than 0.1 nW cm$^{-2}$ sr$^{-1}$ are set to be 0.1 nW cm$^{-2}$ sr$^{-1}$; thus, the minimum of $W_i$ is naught. $\overline{W}$ is the average of $W_i$ in a 2°x2.5° grid cell, and we assume $W_i/\overline{W}$ represents the ratio of the surface $NO_2$ concentration at 0.05°x0.05° to that at 2°x2.5°, due to the relationship between VIIRS nighttime lights and TROPOMI $NO_2$ tropospheric VCDs. The ratio multiplies the surface $NO_2$ concentration at 2°x2.5° $C_c$, to obtain the downscaled result $C_{f,i}^{NL-DC}$. This approach is referred as Nighttime-light Downscaling Concentration (NL-DC).

### 3.3 Downscaling emissions

To simulate or forecast surface $SO_2$ and $NO_2$ concentrations at a resolution of 0.25°x0.3125° through the GEOS-Chem model, the posterior emissions at a resolution of 2°x2.5° should be downscaled to fit the model resolution. The prior MIX 2010 emission inventory has a spatial resolution of 0.25°x0.25°, which is slightly finer than 0.25°x0.3125°, and it can be easily processed to fit 0.25°x0.3125° simulations through the HEMCO – the GEOS-Chem emission processing package (Keller et al., 2014). Thus, all the posterior emissions at a resolution of 2°x2.5° are downscaled to 0.25°x0.25°, which are further regridded to 0.25°x0.3125° with HEMCO. We introduce two emission downscaling approaches with prior MIX 0.25°x0.25° emissions and 0.05°x0.05° VIIRS nighttime lights used as spatial proxies. The two methods are titled Downscaling Emissions with MIX (MIX-DE) and Downscaling Emissions with Nighttime Light (NL-DE).

### 3.3.1 MIX-DE

We assume fine-resolution prior emission inventories have correct relative spatial distributions at fine scales, but a systemic bias exists at coarse scale. The emission downscaling approach is shown in Eq. (4), where $E_{f,i}^{pri}$ is the $i^{th}$ MIX emission estimate at 0.25°x0.25° resolution in a 2°x2.5° grid cell for year 2010, and $E_c^{post}$ is posterior emissions at 2°x2.5° from Part I, i.e. Wang et al. (2019), and $E_{f,i}^{MIX-DE}$ is the downscaled posterior emissions at 0.25°x0.25° resolution.




$$E_{f,i}^{MIX-DE} = E_c^{post} \times \frac{E_{f,i}^{pri}}{\sum_{i=1}^{80} E_{f,i}^{prior}} \quad (4)$$

### 3.3.2 NL-DE

VIIRS nighttime lights are good proxies for allocating $CO_2$ emissions (Ou et al., 2015), and they are also expected to be useful for downscaling $NO_x$ emissions. Figure 3 shows that VIIRS nighttime lights and MIX $NO_x$ emissions

have similar spatial patterns and the linear correlation coefficient between them is as high as 0.73. Thus, VIIRS nighttime lights at a resolution of 0.05°x0.05° are used to downscale $NO_x$ emissions as shown in Eq. (5). $E_c^{post}$ is posterior emissions at 2°x2.5° from Part I (Wang et al. (2019), and $A_i$ and $V_i$ are area and VIIRS nighttime lights at 0.05°x0.05°, respectively. $E_{f,i}^{NL-DE}$ is the downscaled posterior $NO_x$ emissions at 0.05°x0.05°, which is further aggregated to 0.25°x0.25°.


$$E_{f,i}^{NL-DE} = E_c^{post} \times \frac{A_i V_i}{\sum_{i=1}^{2000} A_i V_i} \quad (5)$$

### 3.4 Design of experiments

### 3.4.1 Simulations for October 2013

A set of GEOS-Chem simulation experiments are designed to illustrate the impacts of model resolutions and emission inventories on simulating surface $SO_2$ and $NO_2$ concentrations over China for October 2013, as

summarized in Table 1. Simulations of surface $SO_2$ and $NO_2$ concentrations are validated with in situ observations. C-PRI and C-POS are desiged to show the CGS effect of surface $SO_2$ and $NO_2$ concentrations in coarse (C) - resolution simulations with prior (PRI) and posterior (POS) emissions, respectively. Both C-PRI and C-POS have a simulation resolution of 2°x2.5°, and use the prior and posterior emissions, respectively. MIX-DDC-PRI, MIX-DDC-POS, NL-DC-PRI, and NL-DC-POS illustrate alleviation of the CGS effect through downscaling of surface

concentrations. In MIX-DDC-PRI and MIX-DDC-POS, surface $SO_2$ and $NO_2$ concentrations at 2°x2.5° from C-PRI and C-POS are downscaled to the resolution of 0.25°x0.3125° through the MIX-DDC approach. NL-DC-PRI, and NL-DC-POS downscale $NO_2$ concentrations at 2°x2.5° from C-PRI and C-POS to the resolution of 0.05°x0.05° through the NL-DC approach. JOINT-F-POS is designed to show the impacts of using posterior emissions from joint (JOINT) assimilations on surface $SO_2$ and $NO_2$ forecast at fine (F) spatial scale. In JOINT-F-POS, posterior

$SO_2$ and $NO_x$ emissions from joint assimilations with various observation balance parameter γ from Part I, i.e.



Wang et al. (2019), are used to simulate surface $SO_2$ and $NO_2$ at 2°x2.5°; this parameter is use to balance the importance of the $SO_2$ and $NO_2$ observational terms in the cost function. The simulated surface $SO_2$ and $NO_2$ concentrations are further downscaled to 0.25°x0.3125° through the MIX-DDC approach and 0.05°x0.05° through the NL-DC approach, respectively. F-PRI, MIX-DE-POS, and NL-DE-POS illustrate the improvements of using

downscaled posterior emissions to simulate surface $SO_2$ and $NO_2$ concentrations. All three simulations have a resolution of 0.25°x0.3125°, but use different emission inventories. F-PRI uses the prior MIX emissions, but MIX-DE-POS and NL-DE-POS use the downscaled posterior emissions. Posterior $SO_2$ emissions downscaled through the MIX-DE approach are used in the two simulations, but posterior $NO_x$ emissions used in MIX-DE-POS and NL-DE-POS are downscaled through the MIX-DE and NL-DE approaches, respectively.

**3.4.2 Forecasts for November 2013**

Wang et al. (2016b) used posterior emissions of the current month to improve air quality forecasts of the next month. We implement a similar approach in this study, but monthly emission variation is also considered. With the assumption that the prior MIX emission inventory has proper monthly variations, posterior MIX-DE and NL-DE emission inventories for November 2013 are obtained by multiplying posterior MIX-DE or NL-DE emission

inventories for October 2013, respectively, by the ratios of prior MIX emissions between November and October 2010. As summarized in Table 2, we design a set of experiments for Air Quality Forecasts (AQF) of surface $SO_2$ and $NO_2$ concentrations at fine resolution over China in November 2013. AQF-PRI uses the prior MIX inventory for November 2010 to forecast surface $SO_2$ and $NO_2$ concentrations of November 2013 at 0.25°x0.3125° while AQF-MIX-DE-POS used the posterior MIX-DE inventory for November 2013. AQF-NL-DE-POS is similar to

AQF-MIX-DE-POS, but the posterior NL-DE inventory for $NO_x$ is used. AQF-MIX-DDC-PRI and AQF-MIX-DDC-POS use the prior MIX for November 2010 and posterior MIX-DE for November 2013 inventories to forecast surface $SO_2$ and $NO_2$ concentrations at 2°x2.5°, which are further downscaled to 0.25°x0.3125° through the MIX-DDC approach. Since $NO_2$ hot spots cannot be captured at 0.25°x0.3125° resolution, the NL-DC approach is also applied to the $NO_2$ forecasts. Thus, AQF-NL-DC-PRI and AQF-NL-DC-POS use the prior MIX

inventory for November 2010 and the posterior MIX-DE inventory for November 2013 to forecast surface $SO_2$ and $NO_2$ concentrations of November 2013 at 2°x2.5°, which are further downscaled to 0.05°x0.05° according to VIIRS nighttime light of October 2013 through NL-DC approach.

**3.5 Evaluation statistics**





We use linear correlation coefficient (R), mean bias (MB), normalized mean bias (NMB), normalized centered root mean square error (NCRMSE) (Wang et. al, 2019), and normalized (NMSE) as measures to evaluate GEOS-Chem $SO_2$ and $NO_2$ surface concentrations with in situ observations. NCRMSE measures the spatial distribution difference between forecasts and in situ observations is similar to root mean squared error, but the impact of bias is removed. NMSE is defined as Eq. (6), where $M_i$ and $O_i$ are the $i^{th}$ GEOS-Chem simulation and in situ observation, respectively, $\overline{O}$ is mean of the observations, and N is number of the observations.

$$NMSE = \frac{\frac{1}{N}\sum_{i=1}^{N}(M_i - O_i)^2}{\frac{1}{N}\sum_{i=1}^{N}(O_i - \bar{O})^2} \quad (6)$$

## 4. Results

### 4.1 CGS and MIX-DDC for SO₂

The CGS effect of surface $SO_2$ concentrations in the coarse-resolution (2°x2.5°) simulations (C-PRI and C-POS experiments) is shown in Fig. 4a-d. The GEOS-Chem 2°x2.5° simulation of every grid cell is the average of surface $SO_2$ at ~5x10$^4$ km$^2$ area, while in situ $SO_2$ observations can only represent average concentrations of much smaller area. Considering that all sites are in cities, where emission sources are located, GEOS-Chem 2°x2.5° simulations are excepted to be smaller than in situ observations. In this study, the NMB is -26.7% (Fig. 4c) in the C-PRI simulation, while the C-POS simulation shows an even stronger negative NMB of bias of -65.3% (Fig. 4d), as the posterior $SO_2$ emission is 35.8% smaller than prior MIX 2010.

To decrease the impact of CGS on surface $SO_2$ simulations, both the prior and posterior GEOS-Chem surface $SO_2$ simulations at 2°x2.5° resolution are downscaled to 0.25°x0.3125° through the MIX-DDC approach (MIX-DDC-PRI and MIX-DDC-POS experiments). The downscaled prior (MIX-DDC-PRI) and posterior (MIX-DDC-POS) GEOS-Chem surface $SO_2$ concentrations at 0.25°x0.3125° are shown in Fig. 4e-h. MIX-DDC-PRI and MIX-DDC-POS $SO_2$ simulations show hot spots of up to 270 μg m$^{-3}$ (Fig. 4e) and 80 μg m$^{-3}$ (Fig. 4f), respectively, compared with the largest value of less than 60 μg m$^{-3}$ (Fig. 4a) and 35 μg m$^{-3}$ (Fig. 4b) in the C-PRI and C-POS simulations, respectively.

MIX-DDC-POS $SO_2$ simulations are in better agreement with in situ observations than MIX-DDC-PRI. The NMSE decreases from 4.63 in MIX-DDC-PRI to 1.50 in MIX-DDC-POS, and the linear correlation coefficient (R) increases from 0.32 to 0.36 (Fig. 4g-h).  The NMB changes, however, from 43.4% to -35.3% (Fig. 3g-h),





which implies CGS effect may not be completely avoided at a resolution of 0.25°x0.3125°. We also separately compare MIX-DDC-PRI and MIX-DDC-POS simulations with in situ observations over provincial capital cities, as SO$_2$ hot spots in smaller cities may still be difficult to be captured by the 0.25°x0.3125° MIX-DDC-PRI and

MIX-DDC-POS simulations. The NMB is 115.0% in the MIX-DDC-PRI simulation and reduces to -5% in the MIX-DDC-POS simulation. Additionally, the MIX-DDC-POS simulation shows better spatial pattern than the MIX-DDC-PRI simulation in terms of NCRMSE, although linear correlation decreases slightly. In spite of the improvement of capturing hot spots in term of NMB using the MIX-DDC approach, we should also notice that the coarse resolution simulations (Fig. 4c-d) have larger linear correlation coefficients and smaller NCRMSEs

than the MIX-DDC simulations. Thus, for SO$_2$ simulations, MIX-DDC helps to capture hot spots, but can make spatial distribution worse than the original coarse resolution simulations.

**4.2 GCS, MIX-DDC and NL-DC for NO$_2$**

NO$_2$ has even a shorter lifetime than SO$_2$, thus the GCS effect also exists in the C-PRI and C-POS simulations. Figure 5a-d shows that almost all in situ NO$_2$ observations are larger than the GEOS-Chem simulations, regardless

of using the prior MIX 2010 (C-PRI) or the posterior (C-POS) NO$_x$ emissions. GEOS-Chem surface NO$_2$ averages from the C-PRI and C-POS simulations, sampled according to in situ observational sites, are 49.2% and 54.5% smaller than average of in situ observations, respectively.

The MIX-DDC approach is also applied to downscale NO$_2$ surface simulations (MIX-DDC-PRI and MIX-DDC-

POS experiments), and the results are validated with in situ observations. The MIX-DDC-POS simulation is better than the MIX-DDC-PRI simulation, although the CGS effect still exists. The NMB is -19.3% and -31.8% for the MIX-DDC-PRI and MIX-DDC-POS simulations, respectively (Fig. 5e-h), which implies that 0.25°x0.3125° may be still too coarse to capture hot spots due to the short lifetime of NO$_2$. The larger negative bias in the MIX-DDC-POS simulation than in the MIX-DDC-PRI also lead to that the former shows large NMSE. Despite the negative

bias, R between observations and the MIX-DDC simulations increases from 0.53 in MIX-DDC-PRI to 0.75 in MIX-DDC-POS, and the NCRMSE reduces from 0.96 to 0.67, which is only slightly larger than 0.64 in the C-POS simulation. Thus MIX-DDC-POS can better capture NO$_2$ hot spots and shows spatial pattern as good as  C-POS.

To further alleviate the CGS effect, we downscale GEOS-Chem surface NO$_2$ simulations at 2°x2.5° to 0.05°x0.05° according to VIIRS nighttime light distributions through the NL-DC approach (NL-DC-PRI and NL-DC-POS



experiments), and the results are evaluated with in situ surface $NO_2$ observations (Fig. 5i-l). The largest GEOS-Chem surface $NO_2$ values are less than 35 µg m$^{-3}$ in both the coarse C-PRI and C-POS simulations (Fig. 5a-b), while they are larger than 100 µg m$^{-3}$ at 0.05°x0.05° in the NL-DC-PRI and NL-DC-POS simulations (Fig. 5i-j).

The Scatter plots of the NL-DC-PRI (Fig. 5k) and NL-DC-POS (Fig. 5l) simulations versus in situ surface $NO_2$ observations show that R increases from 0.61 in the NL-DC-PRI simulation to 0.75 in the NL-DC-POS simulation, and NMSE decreases from 3.69 to 1.80, which is smaller than that in the coarse-resolution simulations and the MIX-DDC downscaled simulation. It suggests that NL-DC has the advantage to downscale surface concentrations (without evoking any CTM simulation and its associated needs of computational resources).


The surface $NO_2$ concentrations used for evaluation are derived from measurements of $(NO_2)_m$ assuming $r_{NO2}$ of 0.83 as stated in Sect. 2.1. Due to the lack of information on $r_{NO2}$, we also test the values in the range of 0.7 to 1.0, and the derived $NO_2$ concentrations are used to validate the NL-DC-PRI and NL-DC-POS simulations at the 0.05°x0.05° resolution. Figure 6 shows that the NL-DC-POS simulation has NMSE in the range of 1 to 4, which

is always better than the NL-DC-PRI simulation with NMSE in the range of 2 to 8.

### 4.3 MIX-DE for SO$_2$ simulations

Instead of downscaling simulation results as shown in Sect. 4.1, we directly simulate surface $SO_2$ concentrations at 0.25°x0.3125° resolution through GEOS-Chem over China in October 2013 using the prior MIX 2010 emissions and the posterior emissions. The posterior $SO_2$ emissions at 2°x2.5° resolution are downscaled to

0.25x0.3125 through the MIX-DE approach. The posterior MIX-DE $SO_2$ emissions are smaller than the prior MIX 2010 $SO_2$ emissions over Northern China, Northern China, and Southwestern China (Fig. 7).

The 0.25°x0.3125° GEOS-Chem simulations of surface $SO_2$ for October 2013 with using the prior MIX (F-PRI experiment) and the posterior MIX-DE (MIX-DE-POS experiment) emission inventories, are shown in Fig. 8.

When validating with all in situ $SO_2$ observations, NMSE decreases from 3.73 in F-PRI to 1.55 in MIX-DE-POS, but bias changes from 15.76 µg m$^{-3}$ to -14.98 µg m$^{-3}$. For the same reason in MIX-DDC-PRI and MIX-DDC-POS assessment in Sect. 4.1, we also focus on provincial capital cities. In this scene, NMSE of the MIX-DDC-POS simulation is 1.85 (Fig. 8d), which is much smaller than 15.07 in the F-PRI simulation (Fig. 8c), but it is slightly larger than 1.76 in the MIX-DDC-POS simulation (Fig. 4h). Moreover, NMB decreases from 101.2% in the F-

PRI simulation to -8.4% in the MIX-DE-POS simulation (Fig. 8). R is 0.23 and 0.14 in F-PRI and MIX-DE-POS, respectively, neither of which is significant at the 95% confidence level.





### 4.4 MIX-DE and NL-DE for NO₂ simulations

Posterior $NO_x$ emissions at 2°x2.5° resolution are downscaled through MIX-DE and NL-DE approaches. Figure 9 shows the prior MIX, posterior MIX-DE, and posterior NL-DE $NO_x$ emissions at 0.25°x0.3125° resolutions.

All three emission inventories show $NO_x$ emission hot spots over metropolises (Fig. 9a-c). Compared with prior MIX, posterior MIX-DE is larger over Hebei province, but smaller over most other areas of the North China Plain and Eastern China (Fig. 9d). As posterior NL-DE emission inventory is downscaled according to the VIIRS nighttime light distribution, the difference (Fig. 9e) between posterior NL-DE and prior MIX and the difference (Fig. 9f) between posterior NL-DE and posterior MIX-DE show scattered positive and negative values.


The three emission inventories are used to simulate surface $NO_2$ concentrations at the 0.25°x0.3125° resolution over China in October 2013, that is F-PRI, MIX-DE-POS, and NL-DE-POS experiments in Table 1. All these simulations are evaluated with in situ $NO_2$ observations (Fig. 10). R increases from 0.46 in F-PRI to 0.61 in MIX-DE-POS and 0.58 in NL-DE-POS, and NCRMSE decreases from 0.95 in F-PRI to 0.81 in MIX-DE-POS and 0.85

in NL-DE-POS (Fig. 10d-f). Both MIX-DE-POS and NL-DE-POS show stronger negative NMB and larger NMSE than F-PRI, which should be caused by the CGS effect. MIX-DE-POS has better R and NCRMSE than NL-DE-POS, which may be attributed to the fact that posterior MIX-DE is downscaled according to the distribution of the prior MIX in which point source locations are known, while all posterior emissions at 2°x2.5° resolution are treated as area sources when downscaling through the NL-DE approach, if we assume that sectoral

ratios do not change between prior and posterior emissions.

### 4.5 Impacts of joint assimilations on surface SO₂ and NO₂ simulations

To evaluate the posterior $SO_2$ and $NO_x$ emissions of joint assimilations with various observation balance parameter γ in from Part I, i.e. Wang et al. (2019), we focus on the sum of NMSE of surface $SO_2$ and $NO_2$ as shown in Fig. 11. The experiment of using the prior MIX $SO_2$ and $NO_x$ emissions has the largest sum of NMSE,

which is followed by the simulation using the prior MIX $SO_2$ emissions and the posterior $NO_x$ emissions from separate assimilation. The sum of NMSE of using the posterior $SO_2$ and $NO_x$ emissions of joint assimilations (JOINT-F-POS) with various observation balance parameter γ (as γ increases, the $NO_2$ species is more emphasized in the cost function) is always smaller than that of the experiment of using the prior MIX $SO_2$ emissions and the posterior $NO_x$ emission from separate assimilation and decrease as γ increases. When γ is 1500 or 2000, the sum

of NMSE of using the posterior $SO_2$ and $NO_x$ emissions of joint assimilations is smaller than that of the experiment





of using the prior MIX $NO_x$ emission and the posterior $SO_2$ emission from separate assimilation, but it equals that of the experiment of using the posterior $SO_2$ and $NO_x$ emissions from separate assimilations.

### 4.6 Application for forecasts

Figure 12 shows evaluations of surface $SO_2$ and $NO_2$ forecasts with in situ observations. AQF-PRI $SO_2$ concentrations are generally larger than in situ observations with MB of 45.07 μg m$^{-3}$ and NMSE of 7.97 (Fig. 12a). The MB and NMSE reduces to -7.12 μg m$^{-3}$ and 1.38 (Fig. 12b), respectively, in AQF-MIX-DE-POS. For surface $NO_2$, NCRMSE and R are 0.76 and 0.65 (Fig. 12c), respectively, in AQF-PRI, and change to 0.75 and 0.66 (Fig. 12d), respectively, in AQF-MIX-DE-POS. The stronger negative NMB and larger NMSE for $NO_2$ in AQF-MIX-DE-POS than that in AQF-PRI is likely attributable to the CGM effect. The CGS effect is eliminated 440 in both the AQF-NL-DC-PRI and AQF-NL-DC-POS, which show positive bias (Fig. 12e,f). In the 0.05°x0.05 forecasts, NMSE decreases from 4.61 in AQF-NL-DC-PRI to 3.43 in AQF-NL-DC-POS, and R increases from 0.38 to 0.42.

All the improvements of forecasts are summarized in the Taylor diagrams (Fig. 13), which includes R, normalized 445 standard deviation (the ratio of forecast standard deviation to in situ observations), NMB, and normalized centered root mean square error (NCRMSE). NCRMSE is shown as the distance between the forecast point and the expected (in situ observation) point. The improvements of forecasting surface $SO_2$ concentrations through MIX-DE and MIX-DDC are comparable (Fig. 13a). For $NO_2$ forecast, MIX-DE show larger improvement than NL-DE and MIX-DDC (Fig. 13b). NL-DC is able to better decrease the Coarse-Grid Smearing effect than MIX-DE, but 450 shows larger normalized centered root mean square error.

### 5. Discussion and conclusions

The posterior $SO_2$ and $NO_x$ emissions at 2°x2.5° resolution constrained by OMPS $SO_2$ and $NO_2$ retrievals through the GEOS-Chem adjoint model (Wang et al., 2019) are expected to improve simulations and forecasts of $SO_2$ and $NO_2$ pollutions, but model simulation at such a coarse resolution fails to capture hot spots over cities due to the 455 CGS effect, which prompts the study and development of downscaling techniques. Here, we introduce several downscaling approaches to obtaining surface $SO_2$ and $NO_2$ concentrations at finer resolution, which are further validated with in situ observations. All these methods are demonstrated through simulations for October 2013 and forecasts for November 2013 over China.





GEOS-Chem 2°x2.5° simulations of surface $SO_2$ and $NO_2$ over China in October 2013 using the prior MIX 2010
emissions and the posterior emissions show negative bias due to the Coarse-Resolution Smearing (CGS) effect.
The coarse-resolution simulations are downscaled to 0.25°x0.3125° resolution according to the distributions of
0.25°x0.3125°simulations based on the prior MIX 2010 emissions (MIX-DDC approach). When comparing with
in situ surface observations, the MIX-DDC posterior $SO_2$ and $NO_2$ simulations show normalized centered root

mean squared error (NCRMSE) is 63.7% and 30.2%, respectively, smaller than the MIX-DDC prior simulations.
Compared with the 2°x2.5° simulations, the downscaled 0.25°x0.3125° simulations alleviate the CGS effect, but
do not avoid it completely. To further decrease the CGS effect for $NO_2$, we further downscale the surface $NO_2$
simulations from 2°x2.5° to 0.05°x0.05° according to VIIRS nighttime light observations, which are strongly
related with TROPOMI $NO_2$ tropospheric VCDs (NL-DC approach). The NL-DC $NO_2$ posterior simulation is

better than the NL-DC prior simulation when compared with in situ observations with NCRMSE decreasing from
1.34 to 1.10 and the MB decreases from 18.30 $\mu g\ m^{-3}$ to 10.29 $\mu g\ m^{-3}$, respectively. In terms of evaluating the
downscaled $SO_2$ and $NO_2$ simulations simultaneously, using posterior $SO_2$ and $NO_x$ emission inventories from
joint assimilation is better than only using one ($SO_2$ or $NO_x$) emission inventory from separate assimilation, and
it is similar to using posterior $SO_2$ and $NO_x$ emission inventories from separate assimilation.


Instead of using prior fine-resolution simulations to downscale posterior coarse-resolution surface $SO_2$ and $NO_2$
concentrations, another approach is downscaling posterior emissions for 0.25°x0.3125° simulations. We
downscale the posterior 2°x2.5° $SO_2$ emissions according to the distributions of fine-resolution prior MIX $SO_2$
emissions (MIX-DE). In the 0.25°x0.3125° simulations, posterior surface $SO_2$ is in better agreement with in situ

observations than the prior. Not only are the fine-resolution prior MIX $NO_x$ emissions used to downscale posterior
2°x2.5° $NO_x$ emissions, we also use VIIRS nighttime light observations as proxies to downscale posterior 2°x2.5°
$NO_x$ emissions (NL-DE approach). All these emissions are used to simulate surface $NO_2$ concentrations, which
are validated with in situ observations. The simulations of using MIX-DE and NL-DE posterior $NO_x$ emissions
show smaller root mean square error and larger linear correlation than the prior simulation. The $NO_2$ simulation

using MIX-DE emissions shows better results than that using NL-DE emissions, which may be owing to all $NO_x$
emissions being treated as area sources in the NL-DE approach while the MIX-DE approach has point source
information, if we assume that sectoral ratios do not change between prior and posterior emissions. We also notice
that using the prior fine-resolution simulations to downscale the posterior coarse-resolution surface $SO_2$ and $NO_2$
concentrations is slightly better than simulations using the downscaled posterior emissions.




To study the feasibility of improving surface $SO_2$ and $NO_2$ predictions, posterior emission inventories of the current month are scaled to the next month according to the monthly variations of prior MIX emission inventory, and are subsequently applied to forecasts of the next month. Here we integrate MIX-DE posterior $SO_2$ and $NO_x$ emission inventories for October 2013 and the monthly scale factors derived from prior MIX emission inventory

to obtain posterior $SO_2$ and $NO_x$ emission inventories for November 2013. These are further used to forecast surface $SO_2$ and $NO_2$ concentrations at 0.25°x0.3125° for November 2013, and the results are better than using prior emissions when validated with in situ observations, although the CGS effect is not completed avoided at this spatial resolution. The forecasts of surface $NO_2$ concentrations at 0.05°x0.05° resolutions through NL-DC can eliminate the CGS effect, and the posterior forecast is also in better agreement with in situ observations than the

prior forecast.

In this study, we show the improvements of GEOS-Chem simulations or forecasts of surface $SO_2$ and $NO_2$ concentrations through posterior emissions constrained by integration of GEOS-Chem adjoint and OMPS observations. In the future, we plan to investigate if the posterior emissions can be applied to other models such

as WRF-Chem and WRF-GC at a spatial resolution finer than the finest grid resolution (0.25°x0.3125°) used in this study. In case of global model of chemistry, it is promising to use nighttime light to downscale $NO_2$ simulations so as to obtain a quick look of $NO_2$ air quality at very fine resolution.

*Author contributions*. YW, JW, and DKH designed the research; YW conducted the research; YW and JW

wrote the paper; DKH contributed to writing; CG participated in the GEOS-Chem simulation; MZ prepared VIIRS nighttime light data; WW prepared in situ observations.

*Competing interests*. The authors declare that they have no conflict of interest.

*Acknowledgements*. This research is supported by the National Aeronautics and Space Administration (NASA) through ACMAP program (grant number NNX17AF77G and grant 80NSSC19K0950) managed by Richard Eckman, and through TEMPO project as part of NASA's Earth Venture program (grant number SV7-87011 subcontracted from Harvard Smithsonian Observatory to the University of Iowa). We acknowledge the computational support from the High-Performance Computing group at The University of Iowa.



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



**Table 1. Design of experiments for simulating surface SO₂ and NO₂ concentrations over China in October 2013.**

| Experimental name[a] | Model resolution | SO₂ emissions | NOₓ emissions | Downscaling sfc. SO₂ conc. | Downscaling sfc. NO₂ conc. | Sfc. SO₂ resolution | Sfc. NO₂ resolution |
|---|---|---|---|---|---|---|---|
| C-PRI | 2°x2.5° | Prior MIX | Prior MIX | No | No | 2°x2.5° | 2°x2.5° |
| C-POS | 2°x2.5° | Post | Post | No | No | 2°x2.5° | 2°x2.5° |
| MIX-DDC-PRI | 2°x2.5° | Prior MIX | Prior MIX | MIX-DDC | MIX-DDC | 0.25°x0.3125° | 0.25°x0.3125° |
| MIX-DDC-POS | 2°x2.5° | Post | Post | MIX-DDC | MIX-DDC | 0.25°x0.3125° | 0.25°x0.3125° |
| NL-DC-PRI | 2°x2.5° | Prior MIX | Prior MIX | No | NL-DC | NA | 0.05°x0.05° |
| NL-DC-POS | 2°x2.5° | Post | Post | No | NL-DC | NA | 0.05°x0.05° |
| JOINT-F-POS[b] | 2°x2.5° | Joint post | Joint post | MIX-DDC | NL-DC | 0.25°x0.3125° | 0.05°x0.05° |
| F-PRI | 0.25°x0.3125° | Prior MIX | Prior MIX | No | No | 0.25°x0.3125° | 0.25°x0.3125° |
| MIX-DE-POS | 0.25°x0.3125° | Post MIX-DE | Post MIX-DE | No | No | 0.25°x0.3125° | 0.25°x0.3125° |
| NL-DE-POS | 0.25°x0.3125° | Post MIX-DE | Post NL-DE | No | No | NA | 0.25°x0.3125° |

[a]The nomenclature of the experimental name is as follows. C represents coarse resolution, F fine resolution, PRI prior, POS posterior, DDC dynamic downscaling concentration, DC downscaling concentration, NL nighttime light, MIX prior MIX emission inventory, DE downscaling emissions, JOINT emission inventory from joint inverse modeling.

[b]JOINT-F-POS is a set of experiments of using posterior emission inventories from joint inversion modeling using different observations balance parameter $\gamma$.





**Table 2. Design of experiments for forecasting surface SO$_2$ and NO$_2$ concentrations over China in November 2013**

| Experimental Name[a] | SO$_2$ and NO$_x$ emissions | Model Resolution | Downscaling sfc. conc. (resolution) |
|---|---|---|---|
| AQF-PRI | Prior MIX for Nov. 2010 | 0.25°x0.3125° | No (0.25°x0.3125°) |
| AQF-MIX-DE-POS | Posterior MIX-DE for Nov. 2013 | 0.25°x0.3125° | No (0.25°x0.3125°) |
| AQF-NL-DE-POS | Posterior MIX-DE of SO$_2$ and NL-DE of NO$_x$ for Nov. 2013 | 0.25°x0.3125° | No (0.25°x0.3125°) |
| AQF-MIX-DDC-PRI | Prior MIX for Nov. 2010 | 2°x2.5° | MIX-DDC (0.25°x0.3125°) |
| AQF-MIX-DDC-POS | Posterior MIX-DE for Nov. 2013 | 2°x2.5° | MIX-DDC (0.25°x0.3125°) |
| AQF-NL-DC-PRI | Prior MIX for Nov. 2010 | 2°x2.5° | NL-DC (0.05°x0.05°) |
| AQF-NL-DC-POS | Posterior MIX-DE for Nov. 2013 | 2°x2.5° | NL-DC (0.05°x0.05°) |

[a]The nomenclature of the experimental name is as follows. AQF represents air quality forecasts, PRI prior, POS posterior, MIX prior MIX emission inventory, NL nighttime light, DE downscaling emissions, DDC dynamic downscaling concentration, DC downscaling concentration.



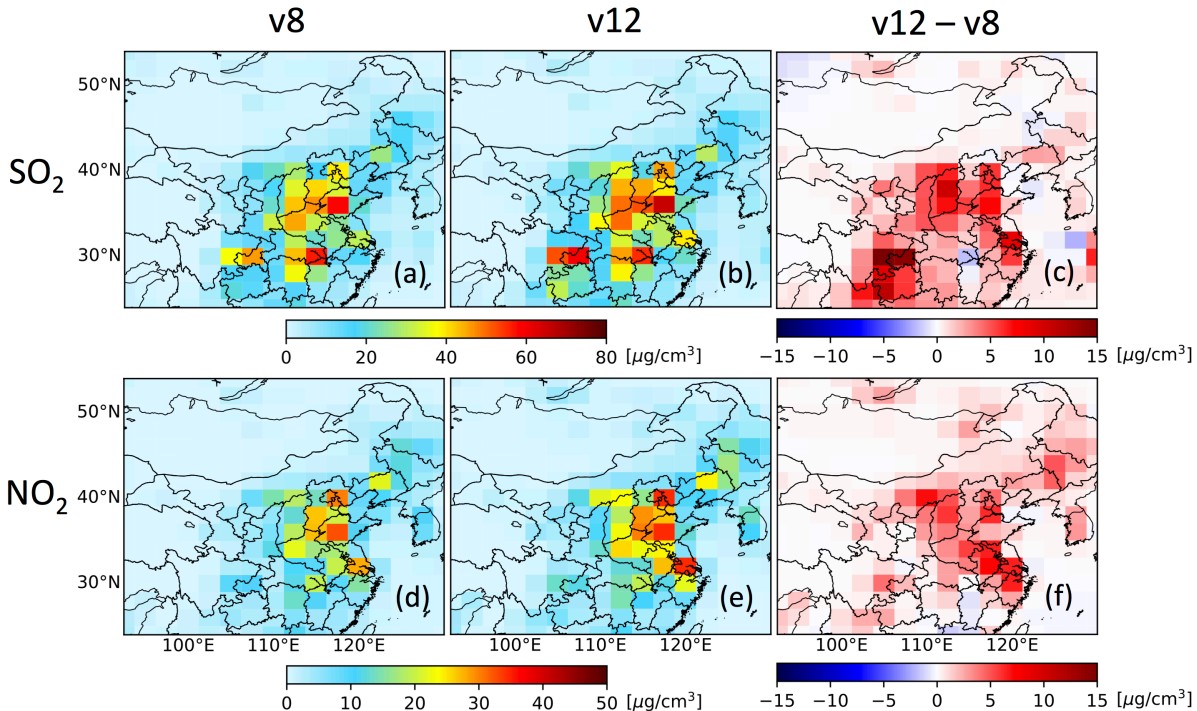

**Figure 1. Simulations of surface SO₂ and NO₂ concentrations for October 2013. (a) and (b) are surface SO₂ simulated by GC-adj v35m (developed based on GEOS-Chem version 8.2.1, updated through version 9, and we name it v8 for short) and GCv12.0.0 (v12 for short), respectively, and (c) is the difference between v12 and v8. (d), (e), and (f) are similar to (a), (b), and (c), but for NO₂.**

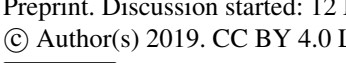



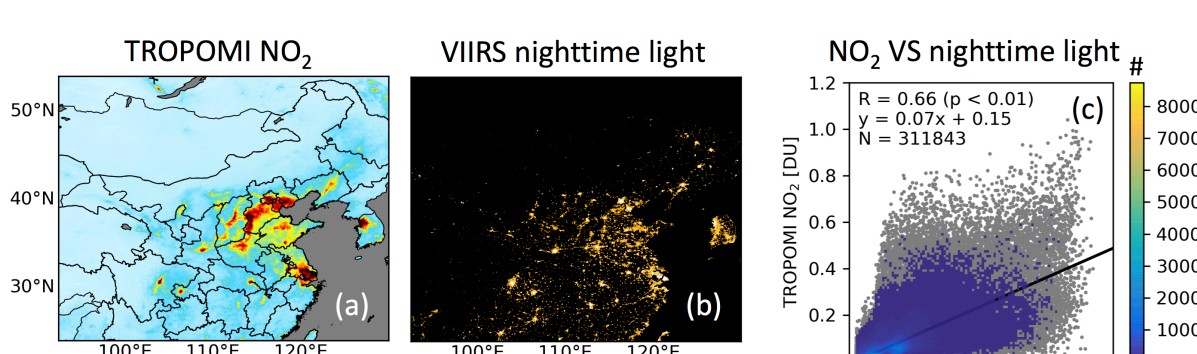

**Figure 2. (a) and (b) are TROPOMI NO₂ VCD and VIIRS nighttime light radiance at the 0.05°x0.05° resolution in April 2018. (c) is a scatter plot of TROPOMI NO₂ versus logarithm of VIIRS nighttime light radiance (grid cells with VIIRS nighttime light radiance less than 0.1 nW cm⁻² sr⁻¹ are removed).**





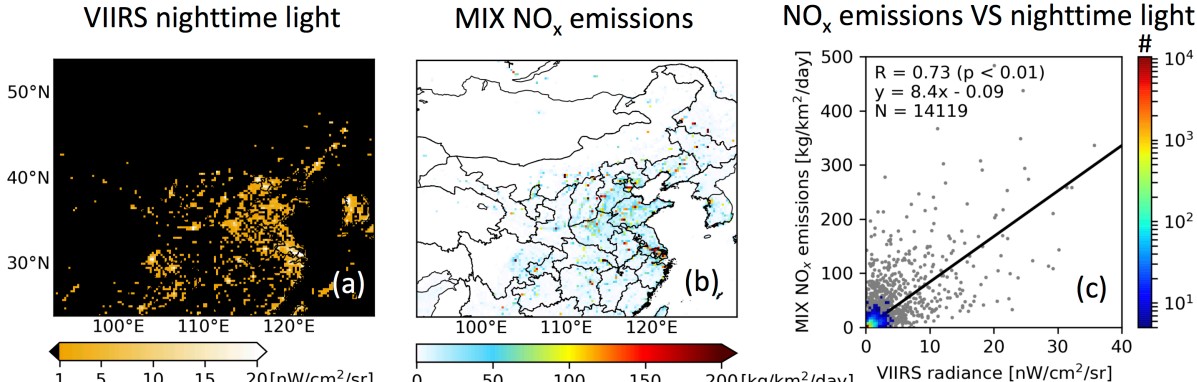

**Figure 3.** (a) and (b) are VIIRS nighttime light radiance and MIX NO$_x$ emissions at 0.25°x0.25° resolution in April 2018 and April 2010, respectively. (c) is scatter plot of MIX NO$_x$ emissions versus VIIRS nighttime light radiance (grid cells with VIIRS nighttime light radiance less than 0.1 nW cm$^{-2}$ sr$^{-1}$ are removed).



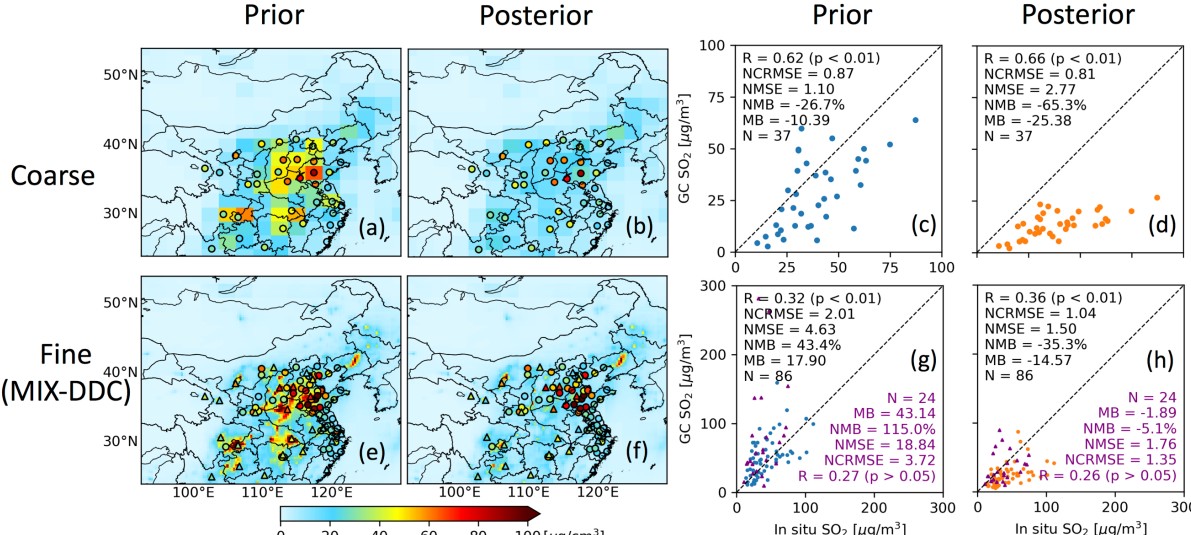

**Figure 4. Evaluations of coarse-resolution, MIX-DDC GEOS-Chem surface SO₂ simulations with in situ observations for October 2013.** (a) and (b) are C-PRI and C-POS simulations, respectively, with in situ observations overlapped. If there are more than one observations in a 2°x2.5° grid cell, they are averaged. (c) and (d) are scatter plots of C-PRI and C-POS simulations versus in situ observations, respectively. (e) and (f) are surface SO₂ concentrations of MIX-DDC-PRI and MIX-DDC-POS, respectively, with in situ province-capital-city (triangle) and non-province-capital-city (circle) observations overlapped. (g) and (h) are scatter plots of MIX-DDC-PRI and MIX-DDC-POS simulations versus in situ province-capital-city (triangle) and non-province-capital-city (circle) observations, respectively. Linear correlation coefficient (R), normalized centered root mean squared error (NCRMSE), normalized mean squared error (NMSE), normalized mean bias (NMB), mean bias (MB), and number of observations (N) are shown over scatter plots, with black color text for all observations and purple color text for province-capital-city observations.



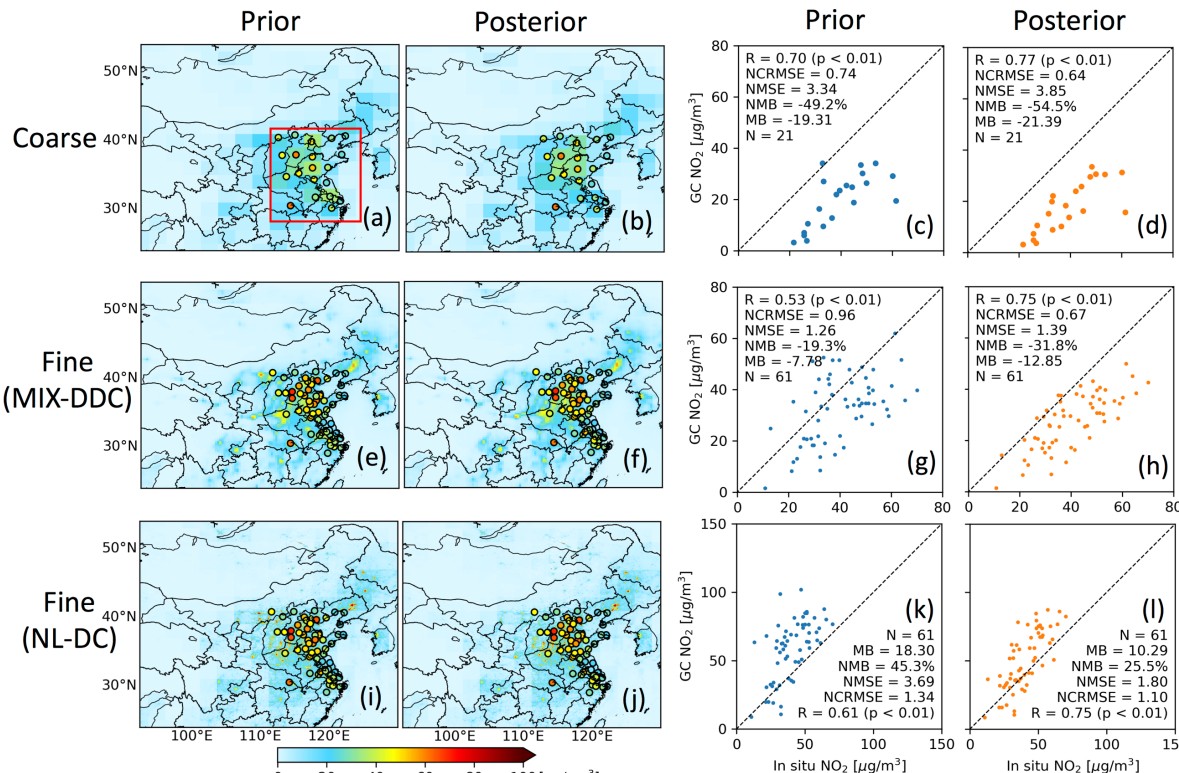

**Figure 5.** Evaluation of coarse-resolution, MIX-DDC, and NL-DC GEOS-Chem surface NO₂ simulations with in situ observations for October 2013. (a) and (b) are C-PRI and C-POS simulations, respectively, with in situ observations overlapped. If there are more than one observations in a 2°x2.5° grid cell, they are averaged. (c) and (d) are scatter plots of C-PRI and C-POS simulations versus in situ observations, respectively. (e) and (f) are GEOS-Chem surface NO₂ of MIX-DDC-PRI and MIX-DDC-POS, respectively, with in situ observations overlapped. (g) and (h) are scatter plots of MIX-DDC-PRI and MIX-DDC-POS simulations versus in situ observations, respectively. (i), (j), (k), and (l) are similar to (e), (f), (g), and (h), respectively, but results are downscaled through the NL-DC approach (NL-DC-PRI and NL-DC-POS). Linear correlation coefficient (R), normalized centered root mean squared error (NCRMSE), normalized mean squared error (NMSE), normalized mean bias (NMB), mean bias (MB), and number of observations (N) are shown over scatter plots. As NOₓ emission is mainly over the North China Plain and Eastern China, validation with in situ surface NO₂ is conducted at these areas (red box in a).





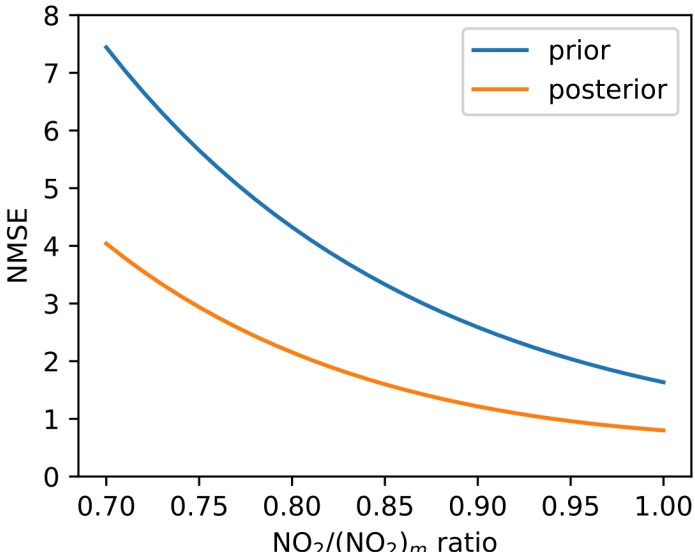

**Figure 6. Normalized mean squared error (NMSE) for NL-DC-PRI (blue line) and NL-DC-POS (orange line) when validating with in situ surface $NO_2$ derived from various $NO_2/(NO_2)_m$ ratio, where $(NO_2)_m$ is measured $NO_z$ concentration.**



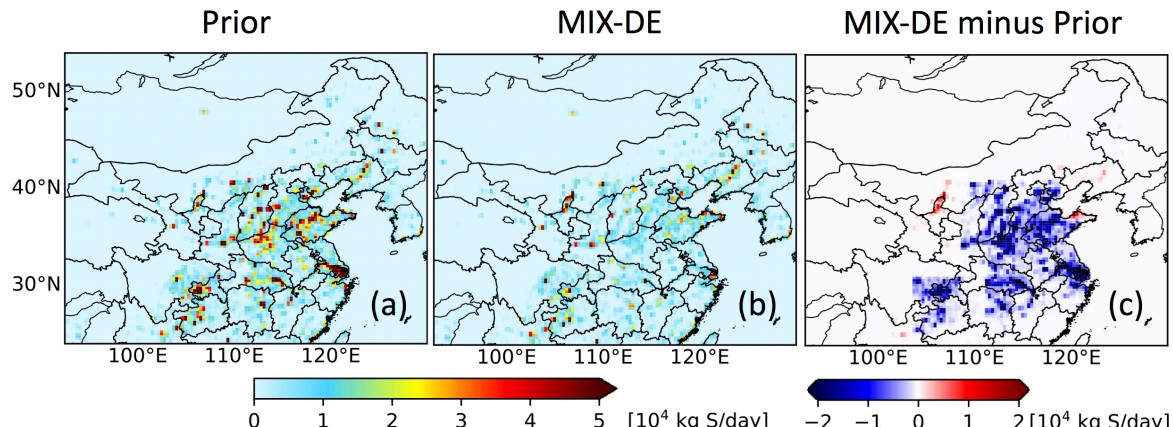

**Figure 7. (a) is SO₂ 0.25°x0.3125° emissions of prior MIX 2010, (b) is posterior MIX-DE, and (c) the difference between posterior MIX-DE and prior MIX 2010.**

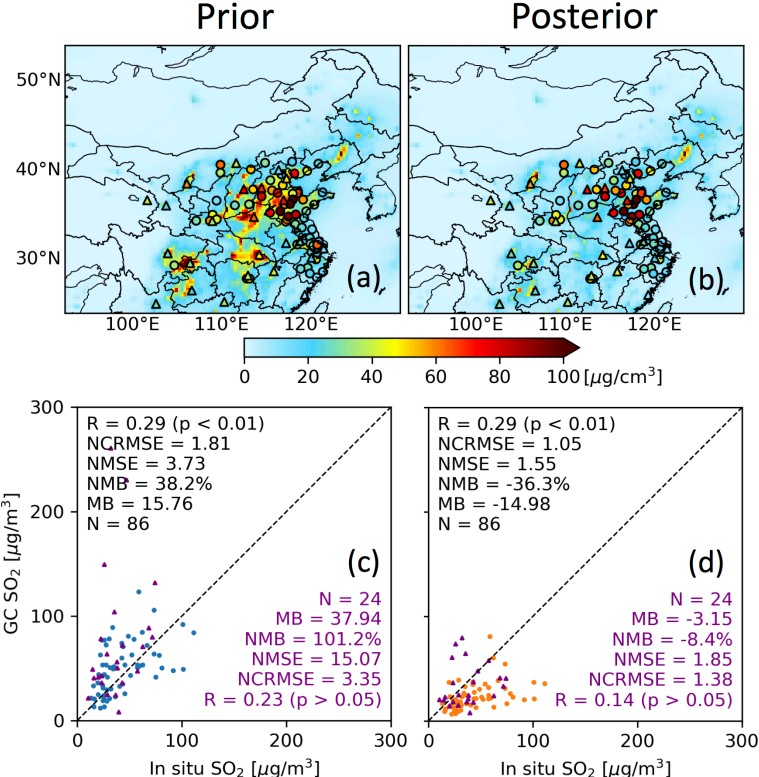

**Figure 8. Evaluations of fine-resolution GEOS-Chem surface SO₂ simulations with in situ observations for October 2013. (a) and (b) are surface SO₂ concentrations of F-PRI and MIX-DE-POS, respectively, with in situ province-capital-city (triangle) and non-province-capital-city (circle) observations overlapped. (c) and (d) are scatter plots of F-PRI and MIX-DE-POS simulations versus in situ province-capital-city (triangle) and non-province-capital-city (circle) observations, respectively. Linear correlation coefficient (R), normalized centered root mean squared error (NCRMSE), normalized mean squared error (NMSE), normalized mean bias (NMB), mean bias (MB), and number of observations (N) are shown over scatter plots, with black color text for all observations and purple color text for province-capital-city observations.**



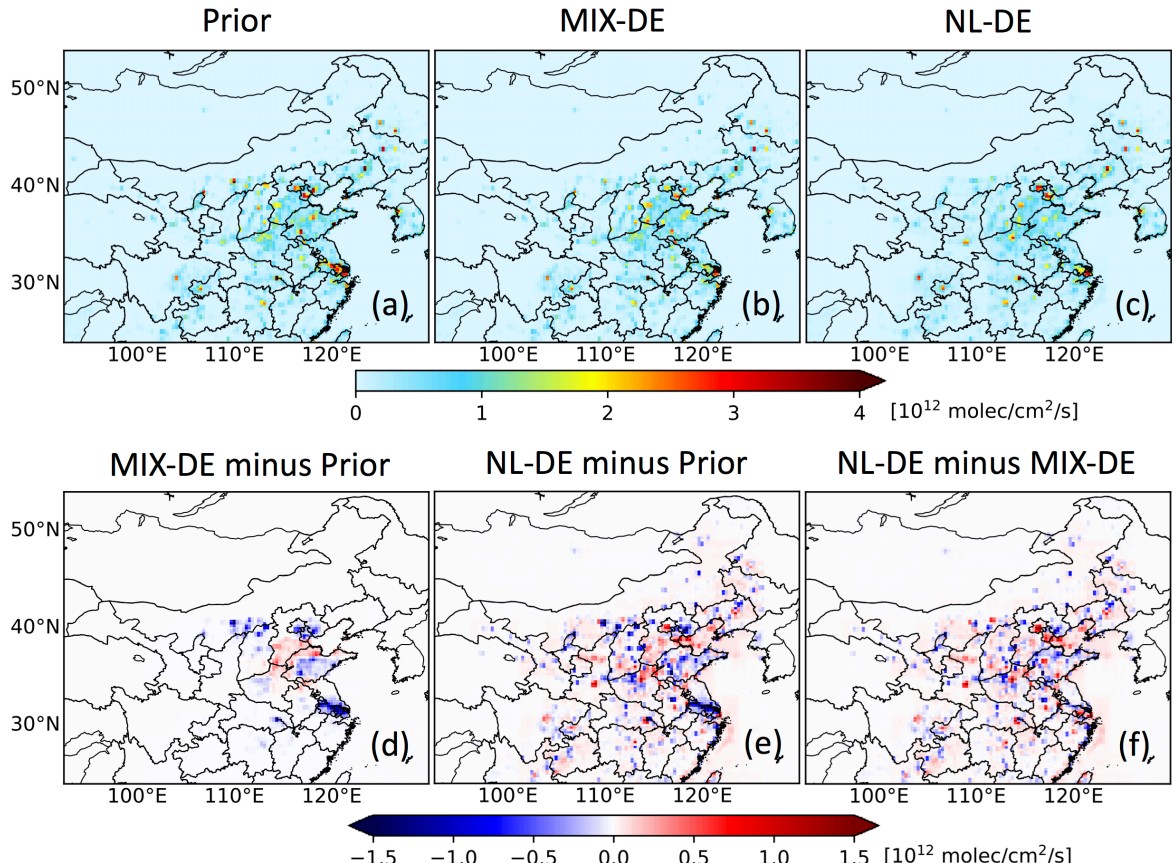

**Figure 9.** NO$_x$ 0.25°x0.3125° emissions of prior MIX 2010 (a), posterior MIX-DE (b), posterior NL-DE (c), the difference between posterior MIX-DE and prior MIX 2010 (d), the difference between posterior NL-DE and prior MIX 2010 (e), and the difference between posterior NL-DE and posterior MIX-DE (f).

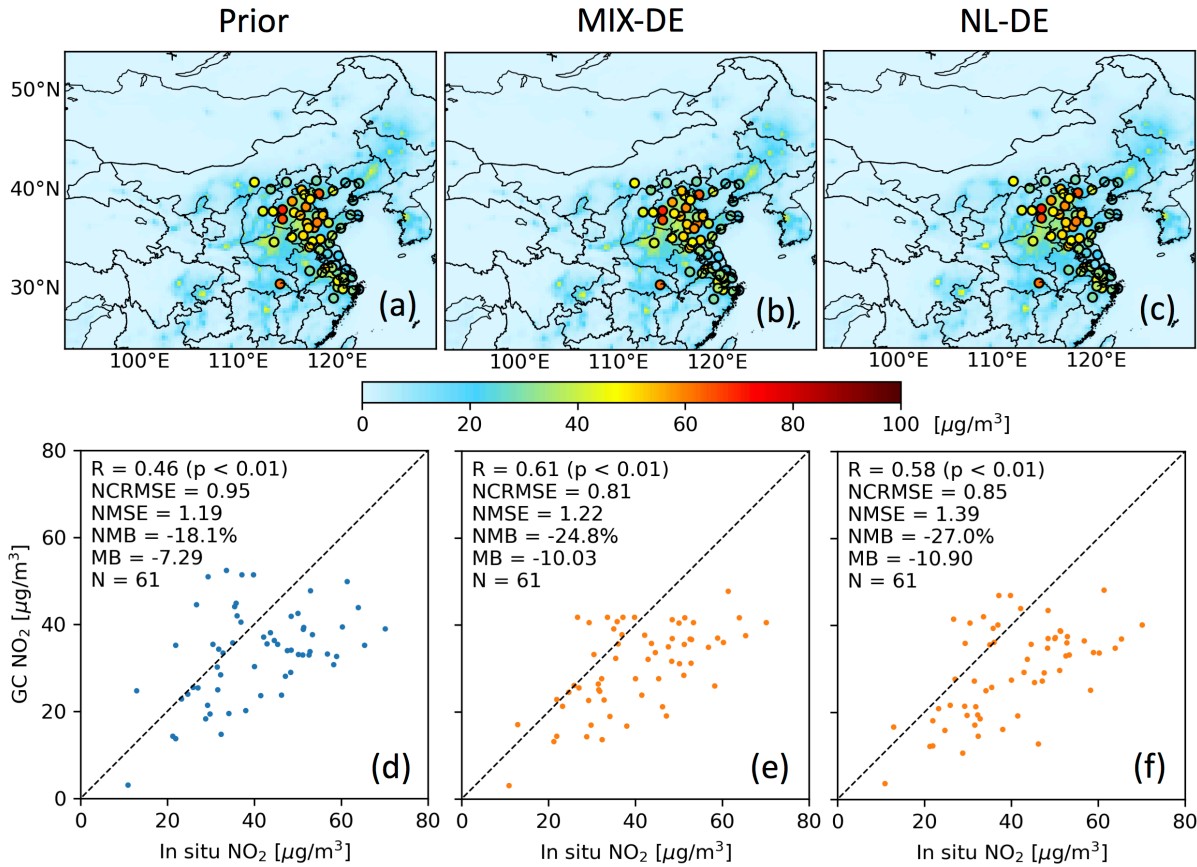

**Figure 10. Evaluations of fine-resolution GEOS-Chem surface NO₂ simulations with in situ observations for October 2013. (a), (b), and (c) are surface NO₂ concentrations of F-PRI, MIX-DE-POS, and NL-DE-POS, respectively, with in situ observations overlapped. (d), (e), and (f) are scatter plots of F-PRI, MIX-DE-POS, and NL-DE-POS simulations versus in situ observations, respectively. Linear correlation coefficient (R), normalized centered root mean squared error (NCRMSE), normalized mean squared error (NMSE), normalized mean bias (NMB), mean bias (MB), and number of observations (N) are shown over scatter plots.**





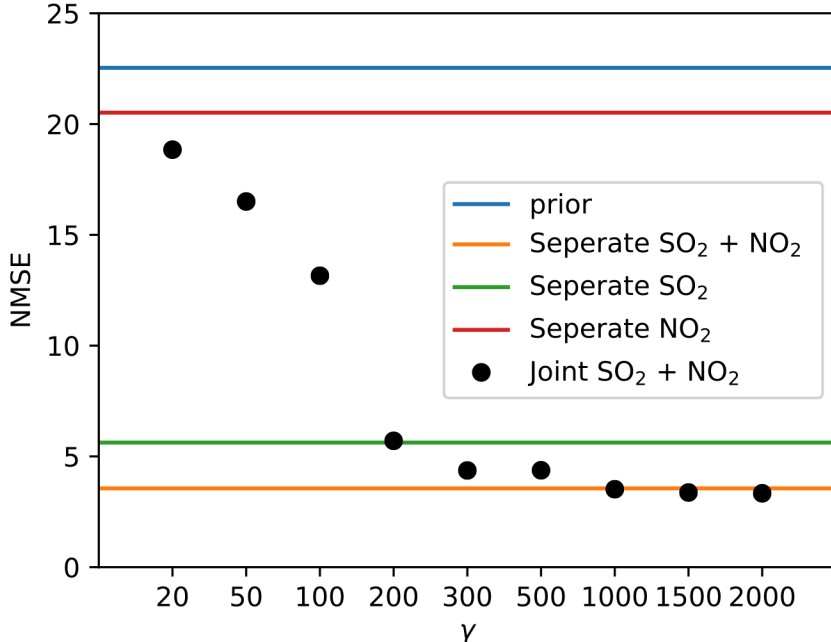

**Figure 11. Sum of normalized mean squared error (NMSE) of surface SO₂ and NO₂. All surface SO₂ and NO₂ simulations come from MIX-DDC and NL-DC, respectively. Black dots are posterior simulations from Joint-F-POS. The blue line is prior simulation results with SO₂ NMSE from MIX-DDC-PRI and NO₂ NMSE from NL-DC-PRI, respectively. The orange line is simulation results with SO₂ NMSE from MIX-DDC-POS and NO₂ NMSE from NL-DC-POS, respectively. The green line is similar to orange line, but posterior SO₂ emission from separate assimilation and prior NOₓ emission are used. The red line is similar to orange line, but posterior NOₓ emission from separate assimilation and prior SO₂ emission are used.**

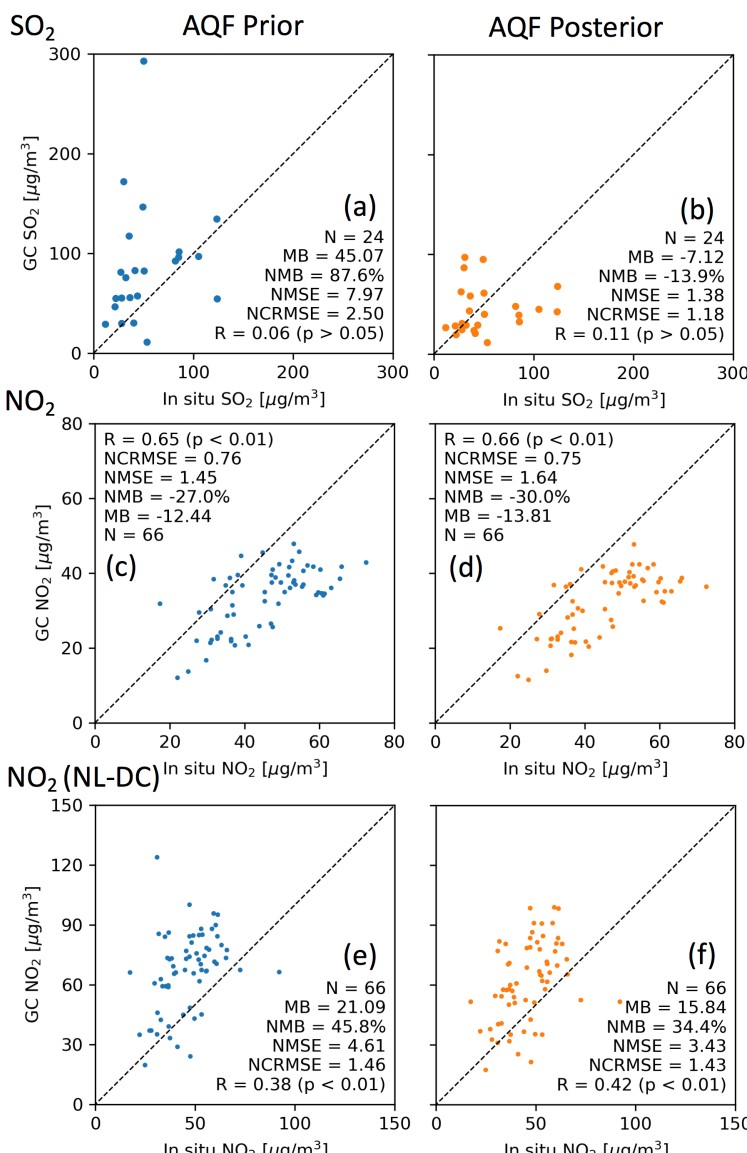

**Figure 12. Evaluation of GEOS-Chem surface SO₂ and NO₂ forecasts with in situ observations for November 2013. (a) and (b) are scatter plots of AQF-PRI and AQF-MIX-DE-POS SO₂ at 0.25°x0.3125° versus in situ province-capital-city observations, respectively. (c) and (d) are scatter plots of AQF-PRI and AQF-MIX-DE-POS NO₂ at 0.25°x0.3125° versus in situ observations, respectively. (e) and (f) are scatter plots of AQF-NL-DC-PRI and AQF-NL-DC-POS NO₂ at 0.05°x0.05° versus in situ observations, respectively. Linear correlation coefficient (R), normalized centered root mean squared error (NCRMSE), normalized mean squared error (NMSE), normalized mean bias (NMB), mean bias (MB), and number of observations (N) are shown over scatter plots.**





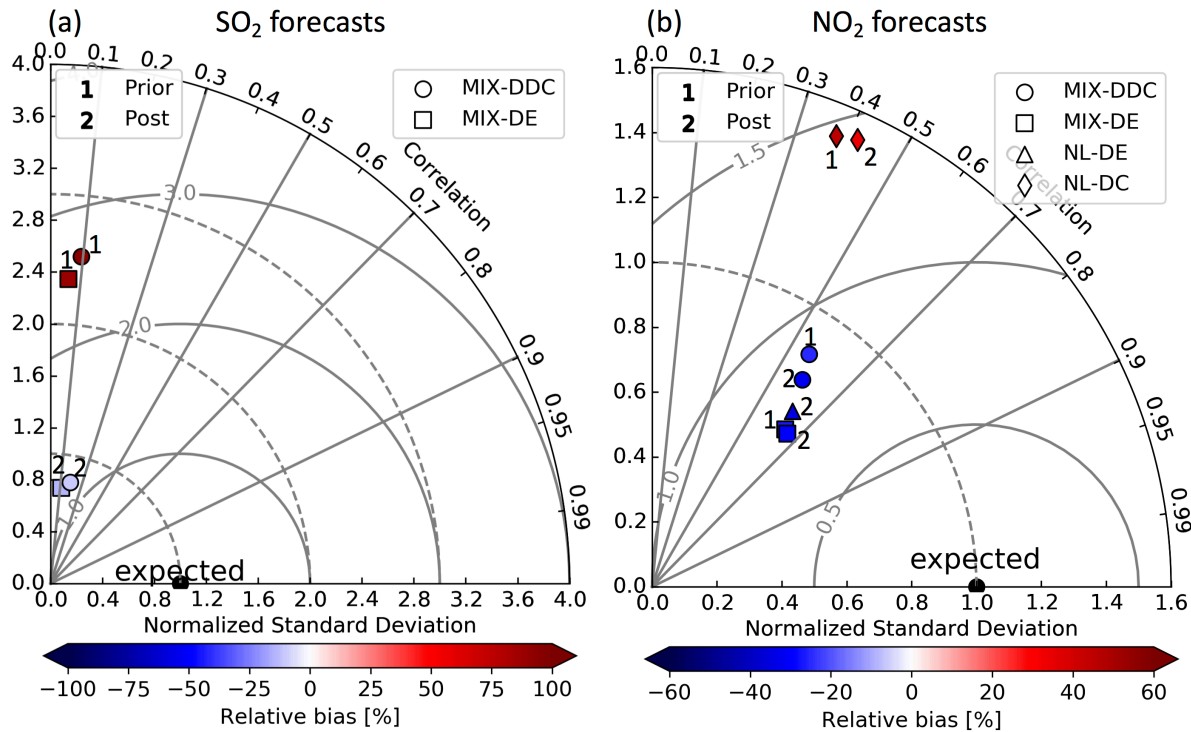

**Figure 13. Taylor diagrams of evaluations of surface SO₂ (a) and NO₂ (b) forecasts with in situ observations. Circle 1 represents AQF-MIX-DDC-PRI, circle 2 AQF-MIX-DDC-POS, square 1 AQF-PRI, square 2 AQF-MIX-DE-POS, triangle 2 AQF-NL-DE-POS, diamond 1 AQF-NL-DC-PRI, and diamond 2 AQF-NL-DC-POS.**