# Peer review of "Inverse modeling of SO2 and NOx emissions over China using multi-sensor satellite data: 2. Downscaling techniques for air quality analysis and forecasts"

_Atmospheric Chemistry and Physics, 2019_

## Referee Comment (RC1) · Anonymous Referee #1 · 11 Dec 2019

General comments:

The paper presented dynamic concentration downscaling and emission downscaling methods for air quality analysis and forecasts. Using the inverse modeling posterior results for October 2013 over China from a companion paper, they applied the downscaling methods to generate both analysis and forecast surface SO2 and NO2 concentrations for November 2013 over China. The results are quite impressive. The paper is well organized, and the overall presentation is very clear.

Specific comments:

Lines 19-20: It is an understatement or even a misleading statement to say that the

joint assimilation of SO2 an NOx is to save computational time.

Line 193: What is the height of the lowest layer?

Lines 297-8: Does "monthly variation(s)" refer to the temporal variation within the month? Please clarify.

Lines 341-2 : Do the authors believe that the negative NMB implies CGS effect? Would 43.4% NMB imply that MIX-DDC-PRI avoided the CGS effect?

Line 351: In what sense is the spatial distribution worse than the original coarse resolution simulations?

Figure 6: How many ratios have been tested here? Showing the actual data points instead of smooth lines will be better.

Figure 11: Can the separate NMSEs of SO2 and NO2 be shown as well? It would be helpful for the readers to understand the model behavior.

Figure 13. "Expected" is misleading as no one would expect the models can achieve such perfect results.

Technical correction:

Line 27: Add "(NL)" after Nighttime light. Line 286: "is use" -> is used.

Line 327: "excepted" -> expected

Line 391: Duplicate "Northern China".

Line 397: MIX-DDC-POS should be MIX-DE-POS.

---

## Referee Comment (RC2) · Anonymous Referee #2 · 2 Jan 2020

This manuscript presents down-scaling results of SO2 and NOx emissions and concentrations based on the coarse-resolution joint emission inversion results from Part 1. The downscaling approaches used information from TROPOMI NO2 observations, MIX inventories, and VIIRS nighttime light observations. The downscaling results were compared against surface in-situ observations. The impact on regional air quality forecasting is also addressed. The prosed approaches are unique and could contribute to improving regional air quality modeling. I would, however, advise the authors to revise the manuscript. These revisions should be made before the manuscript can be considered for publication.

[Figure]

[ Major comments ]

As I suggested for Part 1, all the results need to be revised using higher resolution (at $0.5° \times 0.667°$ degree resolution) joint inversion results for this type of regional study. The $2° \times 2.5°$ resolution inversion could lead to large systematic biases in both local and regional emissions and concentrations in the downscaling analysis, associated with the non-linear chemistry. Ideally, inversion calculations should be done at $0.25°$ or $0.05°$ degree resolution to provide reference information for the downscaling results, but this could be difficult.

It is more straightforward to conduct high-resolution inversions using regional models. There are already several high resolution regional inversion frameworks, for instance, using WRF-Chem. The benefit of using the current coarse resolution global joint inversion framework (rather than regional high resolution inversion systems) to improve regional NOx and SO2 emissions and air quality forecast for China needs to be discussed.

The current manuscript is technical and does not seem provide sufficient scientific implications for ACP (not for GMD). It is required to provide scientific implications based on the proposed approaches. For instance, more detailed information on differences in the spatial patterns between VIIRS nighttime lights and MIX inventories and possible biases in the MIX emission inventories for each emission category would be interesting. Such information will be essential to determine the best downscaling approach for right reason.

The evaluations of forecasts in Section 4.6 are not informative in the current form. Because the purpose of this study is to improve regional air quality forecasts, evaluations of simulated ozone (one of the most important air pollutants) using in-situ observations would add important information.

The use of GCv12.0.0 model instead of GC adjoint v35m could provide some insights into the model dependent posterior emission inventory. Nevertheless, the usefulness

of the proposed downscaling approach should first be evaluated in a consistent framework (GC v35m) to avoid too much complications. Otherwise, it is required to demonstrate the model performance difference in detail.

More specific comments:

3.2.2 I'm wondering if this approach can be applied to SO2. If not, please explain the reason.

L350 "Thus, for SO2. . ." This suggests that the overall spatial pattern was degraded, while capturing hot spots. What emission sources were actually degraded? This would provide important implications into the emission inventories.

L360 "The MIX-DDC-POS. . .". It is not clear to me that the POS is better and the CGS effect still exists (how did you know?).

L365 "Thus MIX-DDC-POS".. Why did the MIX-DDC approach show good spatial pattern for NO2 and not for SO2? The MIX SO2 and NO2 spatial pattern should look similar.

L378 Why are there large positive biases?

L400 The correlation is very low. Please discuss it.

L416 I'm not sure if this is really caused by the CGS effect only. For instance, what happens when posterior emissions are biased?

L417 "which may be attributed. . ." I don't understand the sentence.

———————————————

---

## Author Response (AR1)

**Reply to reviewers and editors:**

We thank all of the reviewers for their careful reading of the manuscript, and for their many constructive feedbacks. The original comments by reviewers are in black font, our replies are in blue.

**Reviewer #1**

*General comments:*

The paper presented dynamic concentration downscaling and emission downscaling methods for air quality analysis and forecasts. Using the inverse modeling posterior results for October 2013 over China from a companion paper, they applied the downscaling methods to generate both analysis and forecast surface SO2 and NO2 concentrations for November 2013 over China. The results are quite impressive. The paper is well organized, and the overall presentation is very clear.

Thanks for the positive comments.

*Specific comments:*

Lines 19-20: It is an understatement or even a misleading statement to say that the joint assimilation of SO2 an NOx is to save computational time.

Thanks for the comment. We want to emphasize emission inventories are initially optimized at coarse resolution. To avoid misleading, we have changed it to "This work thus introduces several approaches to downscaling coarse-resolution (2°×2.5°) posterior $SO_2$ and $NO_x$ emissions for improving air quality assessment and forecasts over China in October 2013. As the Part I of this study, these 2°×2.5° posterior $SO_2$ and $NO_x$ emission inventories are obtained from GEOS-Chem adjoint modeling with the constraints of OMPS $SO_2$ and $NO_2$ products retrieved at $50 \times 50$ km² at nadir and $\sim 190 \times 50$ km² at the edge of ground track."

Line 193: What is the height of the lowest layer?

The height of the lowest layer is in the range of 115 m to 135 m, as shown in the figure below. We have added information to the manuscript and this figure to supplement.

[Figure]

**Figure S1. The box height of the lowest layer of GEOS-Chem in October 2013.**

Lines 297-8: Does "monthly variation(s)" refer to the temporal variation within the month? Please clarify.

Thanks for pointing out this. Here "monthly variation(s)" refers to temporal variations among different months. Temporal variation within a month is not considered. We have added corresponding clarification in the manuscript.

Lines 341-2 : Do the authors believe that the negative NMB implies CGS effect? Would 43.4% NMB imply that MIX-DDC-PRI avoided the CGS effect?

These are good questions. We acknowledge that simulation bias is at least affected by emission bias and the CGS effect. Thus, negative NMB may be CGS effect as well as emission bias. Similarly, 43.4% NMB does not necessarily imply that MIX-DDC-PRI could completely avoid the CGS effect. Compared with 2°x2.5° simulations, 0.25°x0.3125° simulations help to decrease the CGS effect, but it is likely that 0.25°x0.3125° simulations or downscaling 2°x2.5°

simulations to the resolution of 0.25°x0.3125° (such as MIX-DDC-PRI) still cannot completely avoid the CGS effect. Zheng et al. (2017) showed that surface $SO_2$ ($NO_2$) concentration simulations from WRF-CMAQ, when evaluating with in situ observations, have a NMB of -23% (%0), 7% (32%), and 41% (45%) at the resolutions of 36 km (~0.36°), 12 km (~0.12°), and 4 km (~0.04°), respectively, which suggests that (1) the CGS effect and other non-linear resolution-dependent processes can affect the results and (2) these problems are alleviated at the resolution of 0.25°x0.3125°, but are not completely avoided. We have added that CGS effect is only reduced in part, and other factors needs to be investigated (section 4.1 and section 4.2).

Line 351: In what sense is the spatial distribution worse than the original coarse resolution simulations?
We have add "in terms of NCRSME" in the sentence. NCRSME is a good metric for spatial distribution.

Figure 6: How many ratios have been tested here? Showing the actual data points instead of smooth lines will be better.
Thanks for the suggestion. The ratios increase from 0.7 to 1.0 with a step of 0.01. We have replaced Fig. 6 by the figure below.

[Figure]

Figure 11: Can the separate NMSEs of SO2 and NO2 be shown as well? It would be helpful for the readers to understand the model behavior.

Yes. Figure of separate NMSEs of $SO_2$ and $NO_2$ are helpful for the readers to understand the model behavior. In revision, figures below are added the figures to the supplement (Figure S4), with a short description in the main text.

[Figure]

**Figure S4. Normalized mean squared error (NMSE) of surface $SO_2$ (a) and $NO_2$ (b). All surface $SO_2$ and $NO_2$ simulations come from MIX-DDC and NL-DC, respectively. Black dots are posterior simulations from Joint-F-POS. The blue line is prior simulation results with $SO_2$ NMSE from MIX-DDC-PRI and $NO_2$ NMSE from NL-DC-PRI, respectively. The orange line is simulation results with $SO_2$ NMSE from MIX-DDC-POS and $NO_2$ NMSE from NL-DC-POS, respectively. The green line is similar to orange line, but posterior $SO_2$ emission from separate assimilation and prior $NO_x$ emission are used. The red line is similar to orange line, but posterior $NO_x$ emission from separate assimilation and prior $SO_2$ emission are used. In the figure (a), the blue line is covered by the red line, and the orange line is covered by the green line.**

Figure 13. "Expected" is misleading as no one would expect the models can achieve such perfect results.

To avoid misunderstanding, we have replaced "Expected" by "Observation" in the manuscript, as shown below.

[Figure]

**Technical correction:**

Line 27: Add "(NL)" after Nighttime light. Line 286: "is use" -> is used. Line 327: "excepted" -> expected

Corrected.

Line 391: Duplicate "Northern China".

Corrected.

Line 397: MIX-DDC-POS should be MIX-DE-POS.

Corrected.

**Reply to reviewers and editors:**

We thank all of the reviewers for their careful reading of the manuscript, and for their many constructive feedbacks. The original comments by reviewers are in black font, our replies are in blue.

**Reviewer #2**

This manuscript presents down-scaling results of SO2 and NOx emissions and concentrations based on the coarse-resolution joint emission inversion results from Part 1. The downscaling approaches used information from TROPOMI NO2 observations, MIX inventories, and VIIRS nighttime light observations. The downscaling results were compared against surface in-situ observations. The impact on regional air quality forecasting is also addressed. The prosed approaches are unique and could contribute to improving regional air quality modeling. I would, however, advise the authors to revise the manuscript. These revisions should be made before the manuscript can be considered for publication.

Thanks for the positive comments and constructive reviews. We've done our best to address the comments in the revision.

*[ Major comments ]*

As I suggested for Part 1, all the results need to be revised using higher resolution (at 0.5°x0.667° degree resolution) joint inversion results for this type of regional study. The 2°x2.5° resolution inversion could lead to large systematic biases in both local and regional emissions and concentrations in the downscaling analysis, associated with the non-linear chemistry. Ideally, inversion calculations should be done at 0.25° or 0.05° degree resolution to provide reference information for the downscaling results, but this could be difficult.

It is more straightforward to conduct high-resolution inversions using regional models. There are already several high resolution regional inversion frameworks, for instance, using WRF-Chem. The benefit of using the current coarse resolution global joint inversion framework (rather than

regional high resolution inversion systems) to improve regional NOx and SO2 emissions and air quality forecast for China needs to be discussed.

Thanks for the comments. Please see our replies to your comments on part I. The main reason is that the coarser-resolution OMPS data cannot resolve the emission sources smaller than its resolution on a monthly basis. In addition, there are practical issues such as computational cost and the availability of data at fine resolution – the earlier the emissions can be updated by satellite, the better the outcomes of these satellite observations for air quality forecast. The goal of this study is to develop methods potentially for improve forecasting in real time rather than to reconstruct the best historical analysis of high-resolution emissions (that was the goal of (Qu et al., 2019)). Thus, the quality of the forecast is the ultimate test of this study.

While regional model is best suited for air quality forecast, to our knowledge, when this manuscript was prepared, GEOS-chem adjoint model still remains the only CTM that has complete thermodynamic description of the secondary inorganic sulfate-nitrate-ammonium aerosol system and has no need to deal with the issues of chemical boundaries for the regional model. Our understanding is that the full-chemistry 4D-Var is not yet possible in WRF-Chem as its adjoint is made only for GOCART scheme at this point, while CMAQ-adjoint (Zhao et al., 2019) model were in open review after this paper was submitted. Again, treating the chemical boundary conditions in the regional adjoint model need to be further studies. All of this are the reasons that ended up to use GEOS-Chem. We plan to use CMAQ-adjoint in near future.

Zhao, S., Russell, M. G., Hakami, A., Capps, S. L., Turner, M. D., Henze, D. K., Percell, P. B., Resler, J., Shen, H., Russell, A. G., Nenes, A., Pappin, A. J., Napelenok, S. L., Bash, J. O., Fahey, K. M., Carmichael, G. R., Stanier, C. O., and Chai, T.: A Multiphase CMAQ Version 5.0 Adjoint, Geosci. Model Dev. Discuss., https://doi.org/10.5194/gmd-2019-287, in review, 2019.

The current manuscript is technical and does not seem provide sufficient scientific implications for ACP (not for GMD). It is required to provide scientific implications based on the proposed approaches. For instance, more detailed information on differences in the spatial patterns between VIIRS nighttime lights and MIX inventories and possible biases in the MIX emission

inventories for each emission category would be interesting. Such information will be essential to determine the best downscaling approach for right reason.

Thanks for the good suggestion. Please see our reply to your comments for the first part regarding the scientific merit of the paper here. ACP is very broad and methods to improve emission estimates and air quality forecasting are within its scope. Furthermore, we demonstrate the potential application of our method (monthly update of emission at coarse resolution and downscaling it) for regional air quality forecast (for the next month). Finally, following your good suggestion, we have added more discussion to Sect. 4.4. New materials about the relationship among Volatile Organic Compound (VOC), $NO_x$ emission, and $O_3$ air pollutions in Sect 4.6 is also added to make the manuscript have more scientific merits. Below is what we have add to Sect. 4.4.

MIX-DE-POS has improved values of R and NCRMSE than NL-DE-POS; here we discuss the possible reasons and propose future works to improve NL-DE. MIX is a mosaic bottom-up emission inventory, and it is actually the MEIC emission inventory for $NO_x$ emissions over China (Li et al., 2017). The MIX (or MEIC) $NO_x$ emission inventory over China consists of emissions from four sectors including coal-fired power plant, industrial, transport, and residential sectors. Coal-fired power plant emissions in MEIC are derived through extensively using detailed information (including locations of individual units) of 7657 generation units in China (Liu et al., 2015); coal-fired power plant emissions can be accurately placed to grids according to source location information (Li et al., 2017). Thus, if we can allocate posterior total anthropogenic $NO_x$ emissions into the four sectors, we expected that it is better to use the MIX coal-fired power plant $NO_x$ emission inventory rather than nighttime lights to downscale the posterior coal-fired power plant $NO_x$ emissions. For the other sectors in MIX (or MEIC) over China, population density is used to allocate industrial and residential emissions to grids (Li et al., 2017), and transport emissions are distributed according to road networks (Li et al., 2017). Using population density to downscale industrial and residential $NO_x$ could underestimate emissions over urban region, compared with the approach of using nighttime light which could better represent economic development levels (Geng et al., 2017). Whether it is better to use road networks or nighttime lights to downscale $NO_x$ emissions from the transport sector requires future investigations. In this study, the posterior $NO_x$ emission inventory to be downscaled is

total anthropogenic $NO_x$ emissions, which is not allocated into different source sectors. Thus, if we assume that the ratios of every sectoral emissions to total anthropogenic emissions do not change between prior and posterior emission inventories, MIX-DE has an advantage for coal-fired power sector, while NL-DE could benefit the downscaling for the industrial and residential sectors. In future work, we could optimize sectoral emissions rather than total anthropogenic emissions, and subsequently downscale posterior coal-fired power emissions through prior MIX coal-fired power emissions, and ultimately use VIIRS night time light data to downscale posterior industrial and residential emissions.

The evaluations of forecasts in Section 4.6 are not informative in the current form. Because the purpose of this study is to improve regional air quality forecasts, evaluations of simulated ozone (one of the most important air pollutants) using in-situ observations would add important information.

Both $SO_2$ and $NO_2$ are criteria pollutants in the atmosphere defined by US EPA and China, thus we think it is appropriate to evaluate the forecasts of two trace gases and state the improvements in regional air quality forecasts. We also agree that evaluations of simulated ozone are very important, and this part is now added into the revision. As shown below, using downscaled posterior emission inventory helps to improve spatial distribution in terms of NCRMSE, and the improvement of NMB depends on region. We have added the text and Fig. 14 below in Sect. 4.6 and Fig. S5 in the supplement.

In addition to the improvement of $SO_2$ and $NO_2$, AQF-MIX-DE-POS enhances on AQF-PRI in the forecast of surface $O_3$ concentrations (Fig. 14). If all $O_3$ in situ observations in the research domain are used for evaluation, a spatial distribution improvement is shown with NCRMSE decreasing from 1.08 for AQF-PRI to 1.05 for AQF-MIX-DE-POS, but NMB changes from -3.1% to 5.0% (Fig. 14c). Indeed, whether bias becomes smaller or larger depends on region. In the North China Plain and Eastern China where $NO_x$ emissions (or $NO_2$ surface concentrations) are large (the black box in Fig. 14a), forecasts of surface $O_3$ concentration are much lower than other regions; and the NMB is -16.7% for AQF-PRI and -6.3% for AQF-MIX-DE-POS with NCRMSE decreasing from 1.20 to 1.16 (Fig. 14c). In this relatively $NO_x$-rich region, the increase of $O_3$ concentration in AQF-MIX-DE-POS is caused by the decrease of $NO_2$

concentrations; the change of $SO_2$ concentrations has negligible impacts on $O_3$ concentrations (Fig. S5). This implies that if Volatile Organic Compound (VOC) concentrations remain constant, emission control of $NO_x$ emissions will exacerbate $O_3$ pollutions. For the region that is out of the black box, although NCRMSE decreases from 0.82 for AQF-PRI to 0.80 for AQF-MIX-DE-POS, NMB increases from 19.0% to 23.3% (Fig. 14c).

[Figure]

**Figure 14. Evaluation of GEOS-Chem surface $O_3$ forecasts with in situ observations for November 2013. (a) is AQF-PRI $O_3$ forecasts with in situ observations overlapped. (b) is the difference between and AQF-MIX-DE-POS and AQF-PRI $O_3$ forecasts (c) is the Taylor diagram of evaluations of surface $O_3$ forecasts in (a) and (b) with in situ observations. . Circles and squares represent the AQF-PRI and AQF-MIX-DE-POS forecasts, respectively. Labels 1, 2, and 3 represent that all sites, only sites that are within the black box in (a), and only sites that are out of the black box in (a) are used for evaluations.**

[Figure]

**Figure S5.** (a) is similar to Fig. 14c, but in the posterior forecasts, the prior MIX NO$_x$ emission inventory and the posterior MIX-DE SO$_2$ emission inventory is used. (b) is similar to Fig. 14c, but in the posterior forecasts, the prior MIX SO$_2$ emission inventory and the posterior MIX-DE NO$_x$ emission inventory is used.

The use of GCv12.0.0 model instead of GC adjoint v35m could provide some insights into the model dependent posterior emission inventory. Nevertheless, the usefulness of the proposed downscaling approach should first be evaluated in a consistent framework (GC v35m) to avoid too much complications. Otherwise, it is required to demonstrate the model performance difference in detail.

Well, the robustness of the emission inventory should be independent of the CTMs. This is the original motive for us to use a different version of GC to assess the value of the optimized emission. Following your suggestion, we also conducted some evaluations in a consistent framework. Fig. S2 and Fig. S3 are similar to Fig. 4 and Fig. 5, respectively, but using the GC adjoint v35m rather than GCv12.0.0. Apparently, both MIX-DDC and NL-DC works when GC adjoint v35m is used for coarse resolution simulation. All conclusions about downscaling through MIX-DDC and NL-DC from analyzing GCv12.0.0 results can also be drawn from GC adjoint v35m results. It is not surprising that when a consistent framework (GC adj v35m) is used for coarse resolution simulation, all downscaled results show better spatial pattern (smaller NCRMSE) than using GCv12.0.0 for coarse resolution simulation. Considering the manuscript

has shown MIX-DE results are similar to MIX-DDC results, we can expect that MIX-DE should also work in a consistent framework (GC adj v35m). We have added the two figures to the supplement and corresponding text to Sect. 4.1 and Sect. 4.2.

[Figure]

**Figure S2. It is similar to Fig. 4, but GC adjoint v35m rather than GCv12.0.0 is used.**

[Figure]

**Figure S3. It is similar to Fig. 5, but GC adjoint v35m rather than GCv12.0.0 is used.**

***More specific comments:***

3.2.2 I'm wondering if this approach can be applied to SO2. If not, please explain the reason. We did not apply this approach to SO2.

We use VIIRS nighttime to downscale $NO_2$ concentrations as there is strong linear correlation between $NO_2$ VCD and nighttime light as shown in Fig 2c. The strong linear correlation is caused by two reasons: (1) nighttime lights are good spatial proxy for allocating $NO_x$ emissions (Geng et al., 2017); and (2) $NO_2$ lifetime is short (several hours), which means the distribution of $NO_2$ concentration hot spots are highly affected by source locations. We do not expect this approach can be used to downscale $SO_2$ concentrations for the two reasons: (1) nighttime lights are not very good spatial proxy for allocating $SO_2$ emissions as $SO_2$ emissions from traffic sector are very small while nighttime lights are strong over rush traffic road; (2) $SO_2$ lifetime is 1-2 days, which is much longer than $NO_2$ lifetime. We have added the explanation to the section.

L350 "Thus, for SO2..." This suggests that the overall spatial pattern was degraded, while capturing hot spots. What emission sources were actually degraded? This would provide important implications into the emission inventories.

Yes, compared with coarse-resolution simulations, the overall spatial patterns of fine-resolution simulations are degraded, although this conclusion is based on the ground-based observation data that are also in coarse resolution as a whole for describing the spatial pattern. The spatial pattern degradation implies that current chemistry transport simulations of surface $SO_2$ concentrations can capture regional spatial pattern (coarse-resolution) well, but it is difficult to simulate local spatial pattern (fine-resolution); the weakness for describing the local spatial pattern simulation suggests the uncertainties of either bottom-up $SO_2$ emission estimates at fine resolution or locally-resolved meteorological fields (Ge et al., 2017), or both. This uncertainty in bottom-up emission inventories can further stem from distributing $SO_2$ emissions at provincial level to fine-resolution grid. We have added the discussion to Sect. 4.1.

Xing et al. (2015) also showed the difficulty of simulating local $SO_2$ pollution. In Xing et al. (2015)'s research, in situ $SO_2$ observations from US-CASTNET and US-AQS were used for evaluation. US-CASTNET sites are mainly located in rural areas to represent regional air pollution, while US-AQS sites are mainly close to urban areas to represent much smaller area

(local air pollution) (Xing et al., 2015). The linear correction coefficients between WRF-CMAQ simulations of surface $SO_2$ concentrations (108 km x 108 km resolution) over the US and in-situ observations were 0.67 and 0.2 when observations from US-CASTNET and US-AQS were used for evaluation, respectively (Xing et al., 2015).

In this study, observational sites are mainly over urban area, and linear correlation coefficients between GEOS-Chem fine-resolution simulations and observations are in the range of from 0.26 to 0.36 (Fig. 4g and h), which is comparable to the value of 0.2 in Xing et al. (2015)'s research. For coarse resolution simulations, the same sites are used for evaluation, but linear correlation coefficients are in the range from 0.62 to 0.66. In the process of evaluating coarse resolution simulations, there are usually several observational sites in a coarse grid box, and observations from these sites are averaged to compare with the simulation of the coarse grid box. The better spatial pattern at coarse resolution also means it is much easier for GEOS-Chem simulations to capture regional spatial pattern of surface $SO_2$ concentrations than local spatial pattern.

L360 "The MIX-DDC-POS. . .". It is not clear to me that the POS is better and the CGS effect still exists (how did you know?).
Thanks for pointing out this. We would like to express that the MIX-DDC-POS simulation is better than the MIX-DDC-PRI simulation **in terms of spatial pattern (NCRMSE),** although the MIX-DDC-POS simulation has larger negative bias than the MIX-DDC-PRI simulation. We partly ascribe the negative bias to the CGS effect.

We acknowledge that simulation bias is at least affected by emission bias and the CGS effect Thus negative NMB may be CGS effect as well as emission bias. Zheng et al. (2017) showed that surface $SO_2$ ($NO_2$) concentration simulations from WRF-CMAQ, when evaluating with in situ observations, have a NMB of -23% (%0), 7% (32%), and 41% (45%) at the resolutions of 36 km (~0.36°), 12 km (~0.12°), and 4 km (~0.04°), respectively, which suggests that (1) the CGS effect and other non-linear resolution-dependent processes can affect the results and (2) these problems are alleviated at the resolution of 0.25°x0.3125°, but are not completely avoided. We have added that CGS effect is only reduced in part, and other factors needs to be investigated (section 4.1 and section 4.2).

L365 "Thus MIX-DDC-POS".. Why did the MIX-DDC approach show good spatial pattern for NO2 and not for SO2? The MIX SO2 and NO2 spatial pattern should look similar.

The performance of the MIX-DDC approach is largely affected by fine-resolution simulations of surface species concentration spatial pattern using prior MIX emission inventory. Spatial patterns of $SO_2$ and $NO_2$ are comparable at coarse resolution. When come to fine-resolution simulations, $SO_2$ spatial pattern degrade much stronger than $NO_2$ spatial pattern. Thus, MIX-DDC approach show good spatial pattern for $NO_2$ but not for $SO_2$. The reason why the simulation of $SO_2$ spatial patter at fine resolution has been discussed for the question "L350 "Thus, for SO2. . ." This suggests that the overall spatial pattern was degraded, while capturing hot spots. What emission sources were actually degraded? This would provide important implications into the emission inventories. ", as shown above.

L378 Why are there large positive biases?

The bias of surface $NO_2$ concentrations are 45.3% and 25.5% for NL-DC-PRI and NL-DC-POS, respectively, which could come from total emission bias as well as the downscaling process through the NL-DC approach. The sites used for validation are mainly over urban region, and we lack sites that are located over rural region to evaluate if positive or negative bias persists over rural region. Thus, we are not able to determine how much of positive bias in NL-DC-PRI and NL-DC-POS is caused by the NL-DC approach. We have added the discussion in Sect. 4.2.

L400 The correlation is very low. Please discuss it.

As we answer the question above for "L350", we have shown that the weakness for describing the local spatial pattern simulation suggests the uncertainties of either bottom-up $SO_2$ emission estimates at fine resolution or locally-resolved meteorological fields (Ge et al., 2017), or both. This uncertainty in bottom-up emission inventories can further stem from distributing $SO_2$ emissions at provincial level to fine-resolution grid. The low correlation here also implies this problem. We have added the following discussion in Sect. 4.3

We also noticed that R is 0.14 in MIX-DE-POS, which is even smaller than 0.23 in F-PRI. Thus, in the simulations, using prior emissions inventories shows better linear correlation than

using posterior emissions inventories. Conversely. in the forecast, using posterior emission inventories (AQF posterior, R=0.11, Fig. 12b) has better linear correlation than use prior emission inventories (AQF prior, R=0.06, Fig. 12a). The contrast may be caused by the fact that linear correlation coefficient is not a robust metric and should be used together with other metrics to evaluate the model.

L416 I'm not sure if this is really caused by the CGS effect only. For instance, what happens when posterior emissions are biased?

Thanks for pointing out this. We acknowledge that simulation bias is at least affected by emission bias and the CGS effect. Thus, negative NMB may be CGS effect as well as emission bias. Zheng et al. (2017) showed that surface $SO_2$ ($NO_2$) concentration simulations from WRF-CMAQ, when evaluating with in situ observations, have a NMB of -23% (%0), 7% (32%), and 41% (45%) at the resolutions of 36 km (~0.36°), 12 km (~0.12°), and 4 km (~0.04°), respectively, which suggests that (1) CGS effect and other non-linear resolution-dependent processes can affect $SO_2$ simulation results and (2) these problems are alleviated at the resolution of 0.25°x0.3125°, but are not completely avoided. We have added the discussion above to Sections 4.1 and 4.2. We also have let this sentence replaced by "which should be partly caused by the CGS effect, although emission bias and other non-linear resolution-dependent processes could play a role."

L417 "which may be attributed. . ." I don't understand the sentence.

We have replaced the sentence by the text below.

[revised manuscript text omitted]